# Optical Tweezers: Phototoxicity and Thermal Stress in Cells and Biomolecules

**DOI:** 10.3390/mi10080507

**Published:** 2019-07-31

**Authors:** Alfonso Blázquez-Castro

**Affiliations:** Department of Physics of Materials, Faculty of Sciences, Autonomous University of Madrid, 28049 Madrid, Spain; alfonso.blazquez@uam.es or singlet763@gmail.com; Tel.: +34-91-497-2455

**Keywords:** optical tweezers, optical trap, photodamage, phototoxicity, photothermal, reactive oxygen species (ROS), singlet oxygen, oxidative stress

## Abstract

For several decades optical tweezers have proven to be an invaluable tool in the study and analysis of myriad biological responses and applications. However, as with every tool, they can have undesirable or damaging effects upon the very sample they are helping to study. In this review the main negative effects of optical tweezers upon biostructures and living systems will be presented. There are three main areas on which the review will focus: linear optical excitation within the tweezers, non-linear photonic effects, and thermal load upon the sampled volume. Additional information is provided on negative mechanical effects of optical traps on biological structures. Strategies to avoid or, at least, minimize these negative effects will be introduced. Finally, all these effects, undesirable for the most, can have positive applications under the right conditions. Some hints in this direction will also be discussed.

## 1. Introduction

The possibility to trap, move and arrange objects in space without establishing direct physical contact is something that has captured the imagination of many people, not just scientists, for a long time. Not very long ago this fell within the realm of the fantastic, the mystical or, later, into science-fiction. However, during the 20th century real physical advancements (in areas such as optics, photonics, plasmonics or acoustics, to name a few) have paved the way for this achievement to come true at last. One of the most successful approaches is optical tweezers. With optical tweezers light really becomes “tangible” and able to grasp, restrain, rotate or move physical objects within certain limits of size, optical properties, etc.

Optical tweezers moved from the realm of the potential to the real world under the insight and guidance of Arthur Ashkin, who has recently been awarded the Nobel Prize in Physics (along with Donna Strickland and Gérard Mourou) in 2018, precisely for this invention, a revolutionizing approach to microscopic manipulation. The seminal paper on optical tweezers (a “radiation pressure accelerator”) was published in 1970 [1]. In it Ashkin provided proof of the manipulation of microscopic particles with focused laser light. The critical point here was that particles with a higher refraction index than the surrounding medium were attracted to the region of largest light intensity (see Figure 1), an unexpected phenomenon for Ashkin in the first place. It was soon reasoned that radiation pressure forces, rising due to light changing its propagation direction because of different refractive indexes at interfaces, acted in such a way as to displace the illuminated object to the region with the largest photon flux [1]. This conclusion meant that it is possible to select a particular spatial location, through intense light focusing, wherein there is a minimum in the potential energy that tends to restore any displacement undergone by an object located in or close to such a location [2,3,4]. The difference in refractive index between the object to be trapped (with higher index) and the surrounding medium is critical for correct optical tweezing, as it involves changes in the propagation of light. This explanation works well for objects in the so-called Mie regime: when the trapped object is larger than the light wavelength employed in the tweezers. For smaller objects the Rayleigh regime, in which the object is equal to or smaller than the light wavelength, dominates. In the Rayleigh regime light-induced dielectrophoretic forces are invoked to explain the optical trap [2,3]. Nevertheless, the physical basis of optical traps falls outside the topic of this review, and excellent works can be consulted for a deeper treatment of these topics (see above cited reviews and references therein).

Somewhat surprisingly, taking into account that the size of biological cells and optically trapped inert particles certainly match over a wide range, it took more than 15 years for optical tweezers to be employed for trapping cells [5,6]. Again, it was Ashkin who led the research that demonstrated the feasibility of trapping living biological material (tobacco mosaic virus, unidentified “rod-like motile bacteria” and *Escherichia coli* bacteria) with optical tweezers. From this point on, the number of publications and biological applications of optical tweezers grew steadily [7,8,9,10,11]. Some very interesting and broad-reaching applications include: micro-surgery at the single cell level [12,13], single cell-trapping and analysis [14,15], 2D and 3D cellular arrangement and structuring [16,17,18], cell-trapping inside living organisms [19,20], and the study of different (e.g., mechanical) properties of trapped biomolecules like motor proteins or DNA [21,22,23]. Recently, several new approaches to optical tweezers have been, or are currently, under development to increase the performance in biological experiments: Raman tweezers-analyzers [24,25], nanoscale trapping [26,27,28], or holographic/photonic crystal tweezers which try to improve multiple object manipulation [29,30,31]. However, a critical phenomenon conspicuously arises in regards to optical trapping of biological structures: damage to the trapped structure, because the large photon densities involved lead to negative photochemical and/or photothermal phenomena.

In this review the mechanisms of photodamage (Section 2) and thermal damage (Section 3) to biological structures and biomolecules which can occur in optical tweezers will be presented. Knowledge about these mechanisms can help plan experiments and understand why some results are obtained. Additionally, some results and theories in regards to mechanical damage on trapped biological structures will be presented (Section 4), as this has probably been a neglected aspect in the past. Strategies and measures to avoid or, in the least, minimize damage will also be presented (Section 5). Finally, in Section 6 some perspectives and perhaps unexpected useful uses for this optical damage will be outlined at the end of the review.

## 2. Photodamage in Optical Tweezers

The development of optical tweezers is intrinsically linked to the use of laser sources. Lasers are necessary to fulfill the requirements of coherence, intensity and directivity in order to implement the optical trap. But the high photon fluxes associated with the optical tweezers can lead to many undesirable side effects. Since the first reports of lasers being used to interact with microscopic biological samples, it has been clear that the biological damage threshold is reached quite easily. Already in 1965 Amy and Storb reported on the spectacular damaging effect upon mitochondria stained with Janus green B by means of a pulsed ruby laser [32]. In the 1970s the group led by Michael W. Berns published several papers on the dramatic effect of multiline (488/514.5 nm) argon or dye lasers in damaging condensed chromosomes in cultured cells [33,34,35]. The relevant aspect here is that no artificial colorant was introduced in the cell system to sensitize to laser light, as opposed to the experiments by Amy and Storb [32]. Thus, the damage was directly ascribed to the laser light. At first Berns et al. considered a one-photon driven photochemical mechanism or a photothermal one to explain the damage [33,34]. Soon, it was shown that the observed damage could be adequately explained as a result of two-photon photochemistry of histone proteins in the chromatin structure [35]. These preliminary observations set the stage for the three types of damage most commonly observed in cell laser microirradiation: linear (one-photon) photochemistry, non-linear (two- or multiple-photon) photochemistry, and photothermal processes. As, technically, optical tweezing can be considered a particular subtype of laser microirradiation, these will be the main damage mechanisms explored in this review and their role in optical tweezers.

In a first approximation the optical trap was implemented with an argon-ion laser emitting at 514.5 nm and power 1–300 mW [5]. However, clear signs of optical damage were observed for the highest light intensities, which prompted Ashkin and collaborators to shift to infrared lasers to continue the biological trapping experiments [6]. It was reasoned (correctly) that visible wavelengths provide more energy per photon than infrared photons, thereby increasing the probability of linear and non-linear optical damage to the sample. In the following both linear (Section 2.1 and Section 2.2) and non-linear (Section 2.3) mechanisms and experimental examples in regards to laser microirradiation and optical tweezers will be presented for the reader to appreciate the assets and drawbacks of different spectral regions, in order to implement an optical tweezers setup. Additionally, a particular type of linear excitation, the direct optical excitation of singlet molecular oxygen (^1^O_2_), will be described in detail (Section 2.2). This deserves particular attention, as oxygen is present in practically every situation in which biological samples are optically trapped, and, coincidentally, the most popular and frequently employed laser wavelengths for optical traps happen to fall within the absorption bands of ground state molecular oxygen (^3^O_2_). This particular oxygen-driven damage has been, for the most, neglected in the literature. Only recently has it started to be considered as an important source of biological damage.

### 2.1. Linear Excitation Photodamage

Linear, or one-photon, photodamage should be, a priori, the most common type of damage, to be expected for biological samples in an optical trap. Linear photodamage is the result of a molecule (a chromophore) absorbing a single photon, thereby producing an electronic excitation in the molecule (see Figure 2a) [36,37]. In contrast, under non-linear excitation, two or more photons reach the molecule within a very short time, adding up their energies to reach some excited state (see Figure 2b). This second kind of excitation will be introduced in Section 2.3. Once excited, a molecule can dissipate this excess of energy through several deactivation pathways: fluorescence, phosphorescence, molecular rearrangement, molecular reaction, molecular fragmentation, electronic energy transfer, or heat production (see Figure 2b). Usually one deactivation pathway dominates the others, depending on the particular microscopic environment. Commonly photochemical reactions alter or destroy the absorbing molecule, or sensitize other secondary molecules to photodamage. In consequence, it is desirable to keep photochemistry at the lowest level when employing optical tweezers.

As it has been long known that biologically relevant molecules tend to absorb light efficiently in the visible (VIS, 700–400 nm) and ultraviolet A (UVA, 400–320 nm) regions, thus this spectral band has been wisely avoided to trap biological samples almost since the beginning of research in the field. Ashkin, as already mentioned, quickly changed his excitation laser from the visible (514.5 nm) [5] to the near-infrared (NIR) at 1.06 µm [6] precisely for this reason. He observed a fast and efficient disruption of trapped organisms under visible light trapping, a phenomenon he later named “opticution”.

In fact, the possibility to precisely damage certain cell structures or regions by means of VIS/UVA laser microirradiation has been largely exploited in the past few decades to study cell responses to damage and repair mechanisms. Berns et al. already made broad use of this in the 1970s to study chromosome-related events in the cell [33,34]. Ultraviolet (UV) (254 nm) microirradiation has been employed to induce contractions in cardiomyocytes [38]. UVA has allowed selective cellular organelle ablation [39] or as a surrogate treatment to study microeffects of ionizing radiation [40]. However, by far, UVA and VIS microirradiation has been employed to assess genetic damage and its repair mechanisms [41,42,43,44,45,46,47,48,49]. Clearly, this wavelength region is biologically deleterious due to one photon absorption. Particularly concerning is the established genetic linear photodamage induction at these wavelengths, more so if the trapped cell is meant to be studied/manipulated and then allowed to proliferate for different reasons (e.g., in vitro fertilization). Even light doses milder than those employed for genetic damage induction have been found to invoke other negative cell reactions, like autophagy [50].

In summary, it is desirable to move to longer wavelengths (NIR) in order to avoid one-photon absorption processes. As a general trend, direct photochemical reactions are more likely the closer to the UVA and high-energy VIS (350–500 nm). Above 500 nm photochemistry does take place, of course, but the indirect sensitization and production of reactive oxygen species (ROS) becomes the dominant damaging mechanism, in particular superoxide (·O_2_^−^) acting as a precursor to H_2_O_2_, and singlet oxygen (^1^O_2_, see Section 2.2 below). The use of NIR light for optical tweezers is now universally acknowledged by the research community employing this tool. Nevertheless, it is convenient to remark that even in the NIR there are some biomolecules capable of one-photon absorption and photochemistry [51]. For example, hemoglobin and its variants display an absorption tail that extends up to ~800–1000 nm (see Figure 3). Sometimes red blood cells are employed as micrometric “handles”, due to their small size and fitness to be optically trapped (see Figure 4a) [45]. As these cells are loaded with hemoglobin, there can be undesired effects due to NIR light absorption (see Figure 4b) [52]. It seems that the effect was not a thermal one. But whatever the red cell disruption was due to direct hemoglobin photochemistry or ROS production was unclear.

For the most part, in any case, the vast majority of the reported biological damage observed under NIR optical tweezing has to do with the direct optical production of ^1^O_2_, non-linear (two-photon) driven photochemistry, or photothermal effects, to be introduced in the following sections. As a summary, Table 1 presents some important examples of one-photon damage in optical tweezers and microirradiation studies, along relevant experimental parameters.

### 2.2. Direct ^1^O_2_ Light Excitation

Over the past three decades there have been systematic reports of damage to cells or biomolecules trapped with NIR optical tweezers where it was difficult to ascribe a photochemical or photothermal mechanisms to explain the observed results. More particularly, the damage observed occurred after exposure to certain NIR regions for which no known chromophore(s) could properly explain the damaging action. However, the mystery resolves to a great extent if the NIR absorption by dissolved molecular oxygen (O_2_) is taken into account. In the following, the main reports and results on the deleterious action of NIR excitation of O_2_ will be presented.

It is true that the absorption coefficients of ^3^O_2_ (ground state O_2_(X^3^Σ_g_^−^)) in the NIR and VIS are very small. Indeed, these direct optical transitions to the first two electronic excited states O_2_(a^1^∆_g_) and O_2_(b^1^Σ_g_^+^) (indicated as ^1^O_2_(a) and ^1^O_2_(b) in the following) from ^3^O_2_ are considered two of the most forbidden optical transitions in Nature [53,54]. However, small is not negligible. It turns out that under the typical experimental conditions found in microirradiation protocols and, in particular, optical tweezing and manipulation the photon intensities, exposure times and oxygen concentrations “conspire” for more than adequate excitation of ^1^O_2_(a) and ^1^O_2_(b) at the sample. The optical absorption spectrum of high-pressure ^3^O_2_ is shown in Figure 5a [55]. Several absorption bands can be appreciated in the UVA-VIS-NIR range (350–1300 nm). Although the presented data corresponds to high-pressure pure oxygen, the same bands can be observed in atmospheric pressure oxygen or dissolved oxygen [56,57]. The physical-chemical environment in which optical excitation takes place modulates the absorption wavelength and coefficient [58,59,60,61]. However, for the purpose of this review, it can be assumed that these changes are minor and not critical if the optical trap is working within one of the absorption wavelength regions shown in Figure 5a. These bands correspond to vibronic transitions of ^3^O_2_ to ^1^O_2_(a) and ^1^O_2_(b), as schematically shown in Figure 5b [62]. There are two kinds of optical transitions shown: monomol and dimol. Monomol refer to photon absorption by a single ^3^O_2_ molecule, pumping it either to ^1^O_2_(a) or ^1^O_2_(b) (wavelengths shown in Figure 5b). Dimol transitions take place when a single photon is absorbed by a momentary [^3^O_2_:^3^O_2_] interacting pair. The available photon energy is shared by the two molecules, which can end in several vibronic levels (Figure 5b). Our discussion in regard to optical tweezers will deal with monomol transitions. Looking at the wavelengths involved in dimol transitions it becomes clear that they are in the VIS, where linear absorption by several biomolecules (e.g, porphyrins or cytochromes) greatly surpass any oxygen absorption (see Section 2.1 above). Monomol transitions of relevance for optical tweezers, on the other hand, are located in the NIR biological window (1270–690 nm; compare Figure 3 and Figure 5). It is important to remark at this point that, although ^1^O_2_(a) and ^1^O_2_(b) present different physical and chemical properties, ^1^O_2_(b) is considered chemically-inactive due to its very fast internal conversion to ^1^O_2_(a) in condensed media [61]. Therefore, any optical excitation of ^3^O_2_ in the NIR can be considered to produce only ^1^O_2_(a) for practical purposes. In consequence, in the following, when singlet oxygen is referred in the text, it is ^1^O_2_(a) the particular excited state alluded to. Relevant examples of reported biological action and damage in those regions will be presented below.

Singlet oxygen is considered a ROS regarding its biological action [63]. It can react with most cellular components and biomolecules, either directly or indirectly, by giving rise to a host of secondary ROS, radicals and reactive chemical intermediates [64,65]. As such, it is considered a damaging chemical compound. Its direct optical generation (not to be confused with its indirect photosensitized production, see Section 2.1 above) is practically unavoidable under typical optical tweezing conditions, if the optical trap is working within one of the ^3^O_2_ absorption bands (see Figure 5). Some measures can be taken in order to avoid such damage and will be discussed further in Section 5. One of the most obvious is to shift the excitation wavelength out of the band, as, ^3^O_2_ absorption bands are quite narrow and weak. Nevertheless, it can be useful to trap *and* produce singlet oxygen at the same time. Undoubtedly, the assessment of the biological action of singlet oxygen is very important. Oxidative stress conditions derived from the optical trapping can be an area of study on itself, under the paradigm of redox biology. Moreover, under this paradigm physiological cellular responses and processes are under a tightly regulated redox control. Therefore, the possibility to study physiological redox responses under optical trapping must be considered in the near future [62,66]. This topic will be briefly elaborated upon in Section 6.

The possibility that singlet oxygen could be involved in the observed photodamaging action of NIR traps was already advanced almost three decades ago. Svoboda et al. (1994) noted that “*longer-term exposure to light at 1064 nm from a Nd:YAG laser produced photodynamic damage to cells, probably by optically pumping singlet molecular oxygen, a toxic free radical*” [9]. They also highlighted that wavelengths around 760 nm seemed quite damaging. Somewhat later, Mohanty et al. studied a large range of wavelengths (308–1064 nm) in order to determine the degree of damage to microirradiated/trapped cells [42]. They also noted a remarkable damage increase around 760 nm. In general, a robust connection between ^3^O_2_ NIR absorption and photodamage in optical traps has not yet been made, despite several reports pointing in this direction. A compilation of the more relevant papers reporting on the involvement of ^1^O_2_ in optical trap photodamage is next presented. To aid the prospective reader to find a particular wavelength region, the papers are grouped according to the ^3^O_2_ NIR absorption bands (1270–690 nm) shown in Figure 5a.

*1270 nm*. As will be shown in Section 3, wavelengths longer than 1250 nm start to be efficiently absorbed by water, leading to vibrational excitation and heat evolution (see Figure 3). Hence, practical optical trapping at 1270 nm has never been considered for this reason. Nevertheless, it has been shown that irradiation at this wavelength also produces singlet oxygen (O_2_(X^3^Σ_g_^−^) → O_2_(a^1^∆_g_) transition, see Figure 5b). Production of singlet oxygen in this way is enough to kill tumor cells in culture [67,68]. Given the dual source of biological damage, thermal and oxidative, the use of the spectral region between 1200 and 1300 nm is strongly discouraged to trap biological samples.

*1060–1064 nm*. This spectral region is very important in the field of optical trapping. The reason: the enormous availability of Nd-based lasers (Nd:YAG, Nd:YVO_4_, Nd-fiber, etc.) which emit around 1060-1064 nm. Precisely because the use of visible light lasers was patently photodamaging, as evidenced from the first approaches to optical traps [5,6], early researchers in the field moved to the NIR, where Nd-based lasers have been in widespread use since long. Thermal issues at this wavelength will be dealt with in Section 3. The optical transition probability for (O_2_(X^3^Σ_g_^−^) → O_2_(a^1^∆_g_) is lower than at 1270 nm, partly because different initial and final vibrational levels are involved. Nevertheless, the transition is still strong enough to produce singlet oxygen with moderate efficiency.

As mentioned before, Svoboda et al. conjectured on singlet oxygen involvement in the observed photodamage at 1064 nm already in 1994 [9]. However, before that (1991) Liang et al. reported on biological damage when manipulating eukaryotic cell chromosomes with 1.06 μm optical tweezers [69]. The damage was explained in terms of thermal load on the sample. However, for the employed CW laser power (60–200 mW) the temperature rise should have been very small (1–2 °C, see Section 3). It is possible that singlet oxygen was produced during the manipulation, which was the real source of the biological damage observed. A few years later, Liu et al. manipulated human spermatozoa or Chinese hamster ovary (CHO) cells in culture with a Nd:YAG laser (1064 nm, CW or Q-switched) [70]. Damage and loss of cell integrity was again observed, particularly when cells were trapped for longer than 2 min. In pulsed mode the damage was more patent. The authors stated that “... *mechanisms other than heating may be responsible for cell damage*”, as the measured temperature increase was very small (<1 °C per 100 mW).

From this point on, and given the availability of other laser systems also emitting in the NIR, there was a systematic effort to determine the less damaging spectral regions for biological optical trapping between 700 and 1100 nm [42,71,72,73], along with detailed analysis of the biological outcomes of 1060–1064 nm exposure [74,75,76,77,78,79,80,81,82,83,84,85].

The wavelength dependence of different biological outcomes, like cell clonal growth or bacterial movement, were studied to determine the most, and least, deleterious regions in the NIR spectrum (see Figure 6). There are three spectral regions in Figure 6a which clearly reduce clonal growth: ~750 nm, ~900 nm, and ~1060 nm [71]. Roughly the same trend is observed for a reduction/abolishment of bacterial movement (Figure 6b) [72]. There is a clear source of damage at ~1060 nm.

The negative effect of light at 1060–1064 nm has been verified in different biological models. Most experiments have been done with eukaryotic cells. CHO cells have been a productive model for the Berns´ group. Apart from the results already mentioned above [70], further studies were done at several wavelengths (700–1064 nm) and published in the same year [71]. A wavelength of 1064 nm was found one of the most effective as to inhibit cell clonal expansion after laser exposure (see also 760–765 nm and 800–1000 nm below). At that time the authors were reluctant to ascribe the results either to thermal or non-linear phenomena. However, no other explanation seemed adequate to properly explain the biological effect. A few years later, Schneckenburger et al. reported the inhibition of CHO cell colonies in culture after 1064 nm exposure [74]. One-photon absorption by water (implicitly pointing to a thermal mechanism) was proposed as the cause of cell inhibition. An interesting result was presented by Mohanty et al. when NC37 human lymphoblasts were microirradiated and then damage to the chromatin evaluated through the comet assay [42]. Exposure to CW 1064 nm clearly introduced severe chromatin damage as compared to controls. Light at 760 nm (CW) was even more damaging in this sense (see 760–765 nm and Figure 13 below). Recently, the initial phase of cell damage, particularly changes of the local cell morphology at the irradiation spot, has been studied in a digital holographic microscope station after 1064 nm irradiation [81]. No significant changes were observed for exposure times shorter than 20 min at 24 °C. However, certain fast changes (seconds) took place in some of the PaTu 8988T (pancreatic tumor) cells when irradiated under the same conditions but at 37 °C. Thus, warmer conditions can prime the plasmatic membrane for damage, either photothermal or photochemical (singlet oxygen). Recently, experiments trapping human spermatozoa with 1064 nm light and then measuring the swimming performance under additional 633 nm exposure have been published [84]. No direct negative results are reported, but the exposure time to the NIR light was limited to 20 s, on concerns that photodamage could occur to the cells.

Complementary experiments have been done with red blood cells trapped with 1064 nm light. Results were obtained by optically trapping erythrocytes with CW 1064 nm light for up to 5 min and, at the same time, collecting biochemical changes through Raman spectroscopy [82]. It was reported that laser powers below 20 mW led to reversible hemoglobin changes, ascribed to deoxygenation. From 20 to 50 mW changes were irreversible and due to hemoglobin aggregation. Authors disregarded mechanical action by the trap on the blood cell, as free hemoglobin displayed the same behavior. The same was considered for thermal damage, given the low powers involved. They proposed non-linear excitation as the source of damage above 20 mW. However, changes in the Raman spectra as a function of light power display quite linear trends. It can be very plausible that hemoglobin aggregation is the result of ^1^O_2_(a) excitation, more so if oxy-hemoglobin is continuously providing ^3^O_2_(X) within the irradiation spot as a consequence of oxygen photo-detachment from the protein. It is quite known that singlet oxygen induces protein cross-linking and aggregation [65] (and references therein). Recently, de Oliveira et al. have studied the action of 1064 nm or 785 nm optical traps on red blood cells elasticity [83]. Very low powers (10 mW) and short times (1–2 min) were employed. For this, thermal mechanisms can be discarded. Loss of cell elasticity was observed at both wavelengths, more marked at 785 nm. It can be that biological response at 1064 nm is due to low dose singlet oxygen production (similar to the previous paper discussed). At 785 nm, one-photon absorption at the very red edge of hemoglobin absorption spectrum can explain the differences observed, with very low level photochemistry taking place. More research in this interesting area of red blood cell trapping is needed, more so because these cells are sometimes employed as “optical handles” to manipulate other elements (see Section 2.1) [45].

Damage evaluation during optical trapping has also been done in unicellular eukaryotic organisms. In this sense, the baker´s yeast (*S. cerevisiae*) has been the most studied. Aabo et al. have published two papers in which cell growth rate and division were measured as a function of laser exposure at 1070 nm [78,80]. The CW powers employed were very low (0.7–2.6 mW) but the exposure times quite long (up to 4 h), in order to assess the possibility of cell survival after long-term trapping and manipulation. As shown in Figure 7, the growth rate of individual control or trapped yeast cells display a different trend. A power of 2.6 mW delays cell growth at all measured times. Even 0.7 mW result in significant growth inhibition from 90 min exposure or longer. This is an important result, as it shows that there is no photodamage threshold, but continued accumulated damage and that care must be taken even for very low optical powers if long trapping times are involved. Thermal damage is not considered the source of damage, but no alternative mechanism is proposed. However, the “*no-threshold*” condition reported fits nicely with a model of singlet oxygen-mediated cellular damage acting at a low rate for a long time [62].

Morphological damage during optical trapping of *S. cerevisiae* has recently been reported [85]. The experimental conditions are somehow the opposite of Aabo et al.: high power 1064 nm (19–95 mW) but shorter trapping times (15 min). To microscopically compare control vs. laser-trapped cells, microfluidic chambers were employed, which allowed the follow up of cell responses and direct visual comparison. Figure 8a shows an example. The cell in the central chamber was optically-trapped for 15 min (power undisclosed). Nevertheless, the light dose was enough to completely halt cell budding for the next 250 min. In Figure 8b responses of particular cells trapped while budding show the deleterious effect of laser light on that process. The significant increase on generation time and population mortality depending on the light power for a fixed exposure time of 15 min is shown in Figure 9a,b, respectively. Individual cell growth is also inhibited by laser exposure (data not shown). Several causes for these were advanced: heating, ROS production, photochemistry. Heating was considered an important factor. Indeed, the authors proposed to substitute light water (H_2_O) for heavy water (D_2_O) in order to reduce the thermal load, as D_2_O absorbs less at 1064 nm. Ironically, such an experiment may, in fact, *enhance* the phototoxicity, as singlet oxygen has a longer lifetime and chemical activity in D_2_O (see Section 5 and Section 6 below for more on D_2_O) [60,61,62].

Another example of photodamage in a trapped microorganism, the microalgae *Trachydiscus minutus*, has also been reported recently [73]. Several wavelengths were assessed for biological damage, from 735 to 1064 nm, all at 25 mW and for 30 s. The shorter the wavelength, the more damage as measured on the photochemical activity of the photosynthetic centers (see 760–765 nm and 800–1000 nm below). In the case of 1064 nm radiation, practically no damage was reported, even increasing the power to 218 mW. Perhaps, being photosynthetic organisms, these algae have already efficient biochemical defense mechanisms to counteract the toxic action of singlet oxygen. This is a common ROS produced during photosynthesis, and so photosynthetic organisms have evolved efficient ways of disposing of it.

Bacterial cells, being in general smaller than eukaryotic cells, should be more prone to manipulation by optical tweezers. Some papers have published the biological responses of bacterial cells to optical trapping, *E. coli* being the most common experimental model. As already mentioned above (see Figure 6b), bacterial movement was quickly abolished after exposure to a 1064 nm optical trap [72]. The response displayed a trend compatible with a one-photon process, with powers in the 100 mW range which should not rise the temperature more than 1 °C. Notoriously, the photodamage practically disappeared under anaerobic conditions, directly pointing to oxygen (and singlet oxygen) involvement in the deleterious action. An alternative approach to look for subtle damage was investigated by Rasmussen et al., in which the internal pH of four kinds of bacteria, *E. coli*, *Listeria monocytogenes*, *Listeria innocua* and *Bacillus subtilis*, was assessed by a pH-sensitive fluorescent probe or a fluorescent protein [77]. Trapping by CW 1064 nm with 6 mW and 60 min was, in general, quite innocuous, although some populations already displayed pH changes. Increasing the power to 18 mW led to immediate pH alterations (see end of Section 3 for more on thermal action and pH changes).

A relevant study on bacterial photodamage was published by Ayano et al. [75]. In it, growth and cell division rates were studied in *E. coli* after laser exposure to determined amounts of time and at different light powers (excitation at 1064 nm). In this way, different “*damage regions*” and thresholds could be plotted against total light dose or energy (power × time). The results are reproduced in Figure 10 for cell growth (Figure 10a) and division (Figure 10b). It can be concluded that negative cell responses depend on the total dose. A larger light flux can be compensated by a shorter exposure time, and vice versa. Authors noted that cell division was more sensitive to optical trapping than cell growth. As to the damage source, nothing certain is concluded. It is advanced that optical trapping at the subcellular-macromolecular level somehow interferes with the metabolism, perhaps slowing it down, which reflects in the observed responses. It would be very revealing to reproduce these experiments under oxygen-controlling/scavenging conditions (see Section 5 below), to determine oxygen’s role in this.

Finally, a paper was published in 2009 which evaluated the photodamage mechanisms on DNA tethered between optically-trapped microspheres (dual trap) [79]. It was shown unambiguously that laser trapping with 1064 nm induced DNA unfolding with a linear dependence on the dose, and that this damage was due to production of singlet oxygen. Reducing pO_2_ or introducing ^1^O_2_ quenchers (e.g., sodium azide or ascorbate) reduced very significantly the unfolding under the same trapping conditions. Although the authors ascribed the production of singlet oxygen to the polystyrene microspheres employed, a clear photosensitizing reaction is far from feasible under those conditions. Much more probable seems the direct 1064 nm photoexcitation of ^3^O_2_(X) to ^1^O_2_(a) inside the optical trap.

*760–765 nm.* This region of the spectrum presents some interpretation problems in regards to photodamage, as it is located close to the red-tail of the absorption spectra of porphyrins and cytochromes. A hint to understand the source of photodamage is provided by the action spectrum itself. In this spectral region ^3^O_2_(X) absorbs within a very narrow “band” of just a few nanometers, roughly from 760 to 765 nm (see Figure 5). Therefore, optical tweezers making use of these particular wavelengths should be avoided. Some examples from the literature are shown in Figure 11 [86,87,88] (see also Figure 6a).

Vorobjev et al. studied the possibility to manipulate mammalian chromosomes (rat kangaroo cells) with optical tweezers inside living cells [86]. It was patent that light at ~760 nm was much more efficient in producing chromosomal aberrations than any other wavelength studied (Figure 11a). Subsequent research on CHO cells also revealed almost complete inhibition of cell cloning at 740–760 nm, even stronger than 1064 nm light, as already mentioned (see Figure 6a) [71]. In those experiments a Ti:sapphire laser was employed to excite cells between 700 and 1000 nm. Note that light powers (88–176 mW) were the same for 700–1000 nm and for 1064 nm. Additional experiments at these wavelengths were carried out at the end of the 1990s. A fast (60 s), complete reduction of CHO cell cloning efficiency was observed after trapping with a CW Ti:sapphire laser at 760 nm [89]. Trapping with the same laser at 800 nm reduced cloning efficiency to 15% but after 1200 s. This numbers account for the extreme cytotoxic action at 760 nm. Additional experiments were done by these authors on the swimming inhibition and killing of human spermatozoa trapped with the laser. At that moment the authors were unable to propose a formal photodamage source that could explain the enormous difference between 760 and 800 nm, although a more efficient two-photon process at 760 nm was mentioned.

The same authors published another paper shortly after, in which it was disclosed that the CW Ti:sapphire laser they were using had, in fact, a mode beating. This could be promoting a partial mode-locking in the laser cavity and, in consequence, there could be some picosecond pulses leaking to the samples. These pulses would lead to non-linear processes which could explain the observed phototoxicity (see Section 2.3) [90]. Again, assessment of cell integrity of trapped spermatozoa showed a marked damaging action at 760 nm, moderate at 750 and 770 nm, and almost negligible at 780 and 800 nm. When the mode beating was inhibited, cell damage was less but still quite pronounced at 760 nm. Additional photodamage proof came from studies on CHO cells trapped with CW 740 or 760 nm [91]. The induction of a “*giant cell*” phenotype after laser trapping (88–176 mW; 20–300 s) was studied as the endpoint of cell proliferative capacity: the cell is alive but unable to proliferate (i.e., it is dead from a population dynamics point of view). Radiation at 760 nm was at least one order of magnitude more efficient in inducing giant cells than light at 740 nm. Following these reports, it became clear to the research community working with optical tweezers that light in the 760–765 nm should not be employed, as no further papers were published.

To reinforce those findings, some photodamage action spectra have been recently published in relation to redox biology and ROS action on cells [88]. Figure 11c,d present the action spectra for HeLa cell death induction (necrotic morphology and propidium iodide uptake) at several wavelengths from 730 to 800 nm. Both intracellular and extracellular irradiations were done. Both show showed a distinct peak at 760–765 nm to induce cell necrosis. Two examples are displayed in Figure 12. Cells were not trapped; rather they were attached to the substrate. However, the microirradiation procedure was very similar to the one routinely employed in optical trapping. Results in this line were obtained by Mohanty et al. when measuring chromatin damage by the comet assay after cell microirradiation [42]. As it can be seen in Figure 13 exposure to 760 nm light was particularly damaging with longer comet tails as compared with 800 nm.

A publication deserves special mention here, as it is probably the only example of the impact of optical tweezers on a multicellular organism (*Caenorhabditis elegans*) [87]. The researchers employed a transgenic model of *C. elegans* in which the promoter of heat shock protein 16 was coupled to the lacZ gene. Under conditions of cellular stress lacZ is then expressed, leading to synthesis of β-galactosidase, the amount of which can be assessed colorimetrically. The worms were optically trapped with wavelengths between 700 and 850 nm emitted by a Ti:sapphire CW laser. The stress-derived gene expression results after optical trapping are shown in Figure 11b. In parallel, the calculated temperature increase for each wavelength under the experimental conditions is also plotted in the graph (see Section 3). The strongest photo-induced stress occurs at 760 nm, followed by 700 nm (probably one-photon excitation of endogenous chromophores) and, much more weakly, at 850 nm. At 810 nm biological stress was negligible. The authors conclude that the spectral region 700–760 nm should be avoided for biological manipulations.

As mentioned in the comments regarding optical trapping at 1064 nm, the microalgae *T. minutus* has been employed as a model to study the photodamage. The experiments included other wavelengths apart from 1064 nm, namely from 735 to 935 nm [73]. Of all wavelengths, 735 nm was by far the most damaging. However, it is unclear if this is because of generation of ^1^O_2_(b) (unlikely, given the narrow effective absorption band at 755–770 nm), or due to photochemistry derived from cellular chromophores (much more likely). In conclusion, there are not that many publications reporting photodamage in the 700–800 nm region, but all agree in that 755–770 nm should be avoided as there is a very robust photodamage mechanism at work at these wavelengths.

*800–1000 nm.* To finish this section some references are discussed on the adequacy of the 800–1000 nm region for optical trapping of living cells/organisms or biomolecules, from the point of view of ^1^O_2_ generation. Even inside this region, there are some reports of photodamage, the source of which also could be the direct optical excitation of ^1^O_2_. Most of these publications have already been introduced. References [42,71,72,73,86,87,88,89,90] present, in one way or another, results concerning the photodamage action spectrum. Some have been reproduced in Figure 6 and Figure 11. Within the 800–1000 nm range, wavelengths between 800 and 850 nm, and 900 to 950 nm seem to be quite harmless. Mirsaidov et al. studied optical traps at these wavelengths for manipulating *E. coli* with minimal biological impact [92]. In particular they explored light at 840, 870, 900 and 930 nm from a CW Ti:sapphire laser. The results are shown in Figure 14. An important result is that an optical trap in which the biological sample is periodically trapped (“*time-sharing*”) by a scanning beam(s) preserves better the studied sample than a continuously trapping tweezer (see Section 5 below).

Light at 900 nm was the least damaging, while 870 and 930 nm was the most. There is an oxygen band at ~920 nm which can in part explain the observed damage at 930 nm (see Figure 5), although it is very weak. A damaging action has also been reported by others at ~900 nm (see Figure 6) [71,72]. A dual response was observed also by Neuman et al., at ~930 nm and ~870 nm [72]. In fact, proof was found that the photodamage was dependent on oxygen being available in the medium. However, there is no oxygen absorption band ~870 nm, the wavelength seems too shifted into the NIR to promote excitation of endogenous biomolecules, and water still absorbs very little, discouraging any thermal argument as an explanation. A hypothesis to explain these results was advanced a decade ago by Zakharov and Thanh that, perhaps, can help to solve the mystery [93,94]. They proposed a “*combined absorption*” of a single photon of ~890 nm by a transitory molecular complex formed by [H_2_O + ^3^O_2_(X)]. The result is the generation of ^1^O_2_(a) and water in a vibrationally excited state. On first thoughts, they idea may seem extravagant, but dimol absorption bands do exist for transient [^3^O_2_(X) + ^3^O_2_(X)] complexes (see Figure 5) [54,57,62]. Hence, it could be of interest to verify this issue. For example, excitation at 870–890 nm in the presence of singlet oxygen quenchers (e.g., sodium azide) or lifetime enhancers (e.g., D_2_O) can provide proof of the generation and involvement of this ROS in this part of the NIR spectrum.

As a final remark, it would be highly advisable to employ light in the 810–860 nm and 940–960 nm bands for any optical tweezers setup, in order to keep photochemistry and singlet oxygen generation at the lowest, and, at the same time, avoiding vibrational excitation of water and heat evolution. A selection of relevant examples of direct one-photon generation of ^1^O_2_ and photodamage is presented in Table 2.

### 2.3. Non-Linear Excitation Photodamage

To take advantage of the optical forces available in optical traps, it is necessary to provide very high light fluxes, usually of the order of >10^25^ photons cm^−2^s^−1^ equivalent to >1 MW cm^−2^ for optical frequencies [1,4,9]. Arguably, these optical intensities are huge in terms of everyday experience. For example, the maximum irradiance of sunlight at midday in the hottest regions of Earth is around 0.1 Wcm^−2^: at least 10 million times less intense than in an optical trap! However, two-photon optical transitions usually require optical intensities of even higher magnitude [36,37]. Therefore, there exists an intensity regime in the NIR between 10^6^ and 10^7^ W cm^−2^ which allows for optical trapping without much interference from non-linear phenomena [12,15]. Frequently, optical tweezers with these features are assembled using a CW NIR-emitting laser, with powers in the range 10–100 mW and focused in the sample with high numerical aperture (>1.0), high magnification microscopic objectives (40× or higher).

When the threshold irradiance is reached, two photons are absorbed in a very short time, behaving as if one photon of double energy (half wavelength) had interacted with the sample (see Figure 2b for a scheme). As it has been discussed in Section 2.1 above, UVA and VIS light is often quite damaging to biological structures and cells. Hence, non-linear optical absorption of NIR photons, 800 nm for example, would behave as if 400 nm photons were being used to make the optical trap, particularly at the focal plane. Following the previous arguments against employing short-wavelength light for such a goal, the researcher should avoid such non-linear processes to take place in order to preserve the studied sample under the best conditions.

There are many examples in the literature in which, deliberately or not, non-linear optical processes occurred in optical traps. Some will serve to present the effects of non-linear absorption in cells and biological substrates. Examples with CW or pulsed nanosecond traps will be introduced first. Then, some examples of pulsed femtosecond traps will be discussed. Finally, some comparisons among the different systems will be presented too.

Several papers already introduced in previous sections present cellular alterations during optical trapping as a result of non-linear optical phenomena. The group of Berns et al. reported several results with CW optical tweezers in the late 1990s, in which non-linear damage was observed or, at least, proposed as a possible source of damage [70,71,90,95,96]. Excitation wavelengths employed were 1064 nm and 750–900 nm, and CW powers in the 20–300 mW range, with variability depending on the laser and paper. The non-linear behavior was proved because the researchers took advantage of the two-photon excitation of several cellular fluorescent probes (propidium iodide, acridine orange, etc.) to check cell status. Figure 15a shows an example taken from one of those experiments [70]. The probes emission intensity displays a square exponential dependence with the excitation power. Under 1064 nm excitation human spermatozoa and CHO cells showed a preserved cellular integrity. However, upon changing to nanosecond pulsed excitation under very similar conditions cell viability was quickly compromised (see Figure 15b,c). As discussed in Section 2.2 it seems that most biological damage observed in those experiments was the result of one-photon excitation of singlet oxygen. However, the existence of one damaging mechanism (singlet oxygen) does not necessarily preclude the action of another (non-linear excitation). In Figure 15d it can be seen that loss of cloning efficiency in CHO cells after 760 nm exposure was greater when laser pulses excited the sample in comparison when measures were taken to cancel those pulses [90]. To similar conclusions arrived other authors [72,74,87]. From the published data it is advisable to work with CW traps instead of pulsed ones working in the nanosecond regime, as they induce biological damage probably as a result of several different mechanisms (non-linear photochemistry, plasma generation, transient photothermal, thermoacoustic) [70,76]. Particular mention deserves the paper by Zhang et al. in which a CW diode laser emitting at 809 nm was employed to trap murine T cells and CHO cells with irradiances of ~10^6^ Wcm^−2^ [96]. This allowed the two-photon excitation of fluorescent viability probes (fluorescence showed a slope of ~2 in relation to excitation power) for cell follow up during trapping. Even at the highest power employed (190 mW) no cell death was reported by the authors for exposure times of 1000 s. Note that the excitation wavelength, 809 nm, is precisely within the safest spectral region as discussed in Section 2.2.

An alternative approach to CW optical traps is to use lasers with pulses in the femtosecond (fs) range. Commonly, these lasers are tunable Ti:sapphire lasers, with pulse lengths of 100–200 fs, average powers in the 10–100 mW and a frequency of pulses of ~80 MHz (reciprocal time ~12.5 ns between pulses). At this very high repetition rate, they behave in “CW-like” mode for many applications, as the irradiated system has a response time much longer than nanoseconds. At the same time, each pulse “packs” very high power and irradiance, frequently in the 10^10^–10^12^ Wcm^−2^. Thus, some non-linear phenomena are readily observed with these lasers (e.g., two-photon luminescence). For these reasons some researchers have employed and compared fs pulsed lasers to CW lasers as sources for optical traps. Biological damage and cell death are, nevertheless, observed with fs lasers. However, there seems to be a very sharp power threshold for damage to set in (which points to non-linear phenomena becoming non-negligible above a certain power). For example, König et al. described safe optical trapping of CHO cells with a fs laser at powers ≤ 1 mW, 50% loss of colony cloning at 2–3 mW, no cell division at ≥6 mW, and total cell destruction above 10 mW [97]. Hopt and Neher documented similar results in bovine adrenal chromaffin cells trapped with a pulsed Ti:sapphire laser (840 nm 190 fs 82 MHz) [98]. Ca^2+^ levels and a degranulation reaction were studied under irradiation. A power of 2.5 mW was considered safe to study those processes as a result of non-linear cell excitation. But powers of 10 mW were considered quite deleterious. It is interesting to compare these results with those of Zhang et al. [96], as similar wavelengths were employed. König et al. described cell death at 10 mW with a fs laser at 800 nm and 8–10 mW; Hopt and Neher found similar results with a fs 840 nm at 10 mW; finally, Zhang et al. reported cell preservation with a CW laser at 809 nm, power of 190 mW and 1000 s exposure time. Thus, fs lasers in the 800–840 nm range seem safe for optical trapping but at average power levels below 5–6 mW.

The damaging mechanisms of these fs pulsed traps can be diverse. First of all, one-photon absorption can lead to undesirable photochemistry. As mentioned repeatedly in this review, the vast majority of optical tweezers make use of NIR wavelengths to avoid this linear photochemistry. However, fs pulsed systems provide enormous light fluxes (e.g., 10^32^ photons cm^−2^ s^−1^ [97]). At these intensities even almost negligible one-photon absorption can have biological consequences. Additionally, these very same high light intensities undoubtedly favor non-linear optical processes, like two-photon absorption followed by photochemistry. To be fair, it is true that this non-linear behavior can have positive applications, like two-photon stimulated fluorescence, which can be an asset under the right conditions [97,98,99].

Given the very short time of each pulse, thermal effects in the so-called stress confinement regime can take place. This condition is fulfilled when the (photo)thermal pulse is shorter than the time it takes for sound (mechanical waves) to traverse the irradiated volume. This is very much the case for fs pulses. Under the stress confinement condition, even relatively small temperature changes can generate large pressures (>MPa) [100]. This can be a source of cell damage (see Section 3 and Section 4 below). In parallel, at the low energies typical of fs optical traps, low-density plasma generation can take place, too. Again, this occurs in the stress confinement regime, which leads to intense pressure pulses and high-energy free electron-driven chemistry [101]. The pressure pulse can lead to occurrence of nano-cavitation, with bubble dimensions of 100–200 nm at threshold fluences [101,102]. These bubbles, smaller than the diffraction limit at optical wavelengths, can induce damage to several cellular structures (membranes, cytoskeleton, organelles, etc.). Due to their very small dimensions and very transient existence the operating researcher can be oblivious of their existence.

Some reports of trap performance and biological damage when comparing pulsed fs and CW laser systems have been published. It seems that CW traps are slightly better than fs traps at creating a potential well where efficient trapping takes place [99,103]. However, non-linear effects, as mentioned, can be advantageous in certain circumstances. Hence, pulsed fs lasers can offer additional interesting features with a very minor decrease in trap efficiency. In any case, particular attention should be paid to average laser power, as magnitudes above 5–6 mW seem deleterious [103]. Subtler chemical changes, detected by Raman spectroscopy, at lower average powers have also been reported by the same group in red blood cells [104]. The conclusion is that cell morphology reveals gross damage, and alterations at the molecular level can take place well before any observable change in cell structure sets in. As it turns out, however, pulsed fs systems can provide an adequate “scalpel” at the subcellular level, given their nanometric precision. An example is presented in Figure 16a, in which yeast cells (*S. cerevisiae*) were first trapped with the 780 nm Ti:sapphire laser operating in continuous mode to minimize damage. Once trapped, a cell was cut by positioning the laser at the cell membrane and switching to fs pulsed mode [105]. Intracellular material was subsequently released to the medium. This approach can have interesting analytical and manipulative applications, as the authors themselves showed. By trapping in CW mode and then switching to fs pulsed mode they were able to trap a yeast cell (CW), cut its membrane locally (fs pulsed), and finally extract an organelle with the optical tweezers (CW). The sequence of events is displayed in Figure 16b.

A couple of other papers in which comparison among different laser systems was discussed deserve mentioning. They do not deal directly with biological alterations in optical traps but the results can be extrapolated to them. Kong et al. compared different types of lasers in order to induce a DNA damage response in HeLa cells [44]. Different wavelengths (337, 405, 532, 800 nm) and laser pulse profiles (CW, ns, ps and fs) were employed. All lasers induced the DNA damage response although with different features (see Section 2.1 above). Mechanisms considered included linear and non-linear optical absorption, generation of low-density plasma, and photothermal effect (see next section). Gassman et al. describe similar results and techniques in their recent review on cellular micro-irradiation [47]. The type and extent of lesions varied depending on such properties as laser pulse timing, wavelength, presence of a photosensitizing compound, etc. The type of genetic lesion was different for different experimental parameters. Finally, Wang et al. published experiments on autophagy induction after laser micro-irradiation [50]. All experimental conditions increased autophagy but the triggering mechanism seemed different. For the CW lasers (473, 543 and 650 nm) ROS are proposed as the causative agent. For the 750 nm fs laser plasma generation inside the cell is more likely the cause. An interesting detail is that Ca^2+^ seems critical for the induction of autophagy with the fs laser but not for the CW ones.

To aid the reader some important examples of non-linear biological damage have been summarized in Table 3 below.

## 3. Thermal Damage and Stress in Optical Tweezers

A significant rise in the local temperature within and around optical traps has been a concern in optical tweezers since the first reports in this research field [6,69,70,106,107] and still is a matter of discussion [12]. Intense absorption in the VIS precluded use of this wavelength range to trap biological samples without damaging them (see Section 1 and Section 2.1) [5,6]. Moving to NIR wavelengths solved this problem to a great extent, but excessive sample heating due to infrared photon absorption (in particular by water molecules) could be a problem. Then, for over three decades, many researchers have devoted their study of optical tweezers to measure the increase in temperature (∆T), its effect on biological molecules and organisms, and to provide measures to counteract the undesirable thermal side effects of microirradiation.

Ashkin et al. advanced in 1987 that, although trapping at 1064 nm had undeniable advantages, using 80 mW would lead to a ∆T “estimated to be several degrees Centigrade.” [6]. As we will see this, although correct in the general direction, was an overestimation of ∆T by an order of magnitude. Articles published a few years later reported experimental values of ∆T as a function of the light power in different cell models: <1.0 °C ± 0.30 per 100 mW (human spermatozoa and CHO cells, 1064 nm [70]); 1.45 °C ± 0.15 per 100 mW (liposomes, 1064 nm [106]); 1.15 °C ± 0.25 per 100 mW (CHO cells, 1064 nm [106]); and 4 °C per 55 mW (water, 985 nm [107]). Check for the corresponding water absorption spectrum in Figure 3. An example of the increase in temperature as a function of laser power is shown in Figure 17a.

Additional measurements of ∆T as a function of average power of the optical trap were published some years later [108]. Authors measured the changes in Brownian movement of trapped (1064 nm) polystyrene and silica microparticles in water and glycerol. From this, the ∆T was calculated. Values are displayed in Figure 17b (from an original table). Differences in solvent viscosity and thermal conductivity lead to ∆T~3–6 K/100 mW (glycerol) and 0.4–1.3 K/100 mW (water). Thus, changes in irradiation parameters and media can modify the thermal properties of the optical trap setup. In this line, it was recently reported by Català et al. that the historically assumed “∆T~1 K/100 mW” rule of thumb for aqueous trapping at 1064 nm can be quite misleading, deserving a warning note. They measured ∆T~2–4 K/100 mW for polystyrene, melamine and silica microparticles trapped with a CW 1064 nm laser [109]. The ∆T seems to depend critically on focusing features like magnification employed, numerical aperture, solvent, etc.

An example of the critical role of the wavelength is shown in Figure 18. Employing the same trapping parameters but using 820 nm or 980 nm (last one close to a water IR resonance band, see Figure 3) leads to diametrically opposite outcomes for cell survival [110]. Jurkat cells show no morphological signs of damage even after 50 min of continuous trapping at 820 nm. In contrast, they display aberrant shapes and damage signs after 10 min at 980 nm. Note that both wavelengths employed do not coincide with oxygen absorption bands (see Section 2.2 and in particular Figure 11b [87]).

In general, it can be concluded that thermal damage, for typical trapping powers, is not the result of reaching hyperthermic intracellular conditions (i.e., T > 42–43 °C). Most papers report ∆T of 1–10 K. For ambient laboratory temperatures of 20–25 °C under optical trapping conditions this means reaching equilibrium temperatures of ~30–35 °C in the trap. The relevant parameter, then, seems to be the *rate of change* of temperature as a function of time (i.e., ∆T/t), and not the absolute final temperature. Clearly it is not the same to reach a final temperature of 35 °C, starting from 20 °C, in 1 s or in 1 ms. Wetzel et al. nicely proved this in their experiments of cell damage (assessed through fibroblast spreading) induced with a CW 1064 nm laser [111]. Although very high temperatures (47–68 °C) were attained, cells showed quite a good viability if the exposure time to the high temperature was short enough. For example, for T_f_ = 48° ± 2 °C an 80–90% cell survival was observed if the laser exposure was 2 s or shorter. Cellular membranes are one of the most vulnerable structures to changes in temperature because of their phase transitions between gel and liquid states [81,112]. Rapid changes in T can induce membrane alterations, compromising cell viability.

Large heating rates affect cellular membranes in two ways: they induce large spatial thermal gradients (intracellular vs. external medium), and they are a source of thermoelastic pulses. Both mechanisms are interrelated and trace their origin to the finite time it takes for heat to flow from the irradiated spot to the surrounding medium. Spatial thermal gradients inside and around optically trapped cells have been shown to lead to damage. For example, it has been reported recently that trapping red blood cells with relatively high powers (*P* > 280 mW) at 1064 nm produces very fast damage and cell collapse [113]. An example is shown in Figure 19. Almost instantaneous collapse of the cell is observed when trapped with 360 mW. The authors estimate spatial thermal gradients of 10^9^–10^10^ Km^−1^, which due to the Soret effect and thermopolarization (see below) translate into transient voltages across the cell membrane of ~1 V. This is enough to induce electroporation and osmotic shock. Remarkably, for trapping powers of 500 mW or higher, the cell collapse is so fast and violent that intense hydrodynamic waves spread out and mechanically damage nearby non-trapped cells.

These fast thermal gradients can lead to thermoelastic mechanical waves [24,44,100]. In [114] the response of a red blood cell to sudden exposure to 100 mW of a CW 1064 nm laser was computationally modelled. Maximal T on the simulations was 32.5 °C but a large part of the temperature rise takes place in a short time, less than 0.01 s (∆T/t > 1000 Ks^−1^). Thermal expansion of the cellular volume engenders mechanical waves that can affect and damage the cell itself and nearby structures (see Section 4). Perhaps this thermoelastic phenomena are behind the observed differences behind the pulsed (time-shared) and CW viability displayed by bacteria in Figure 14 [92]. Furthermore, long-term alterations can be taking place in previously trapped cells. If thermal and thermomechanical phenomena occur, but are below the level of immediate cell destruction, conditions for cell membrane poration can be the right ones for cells to uptake compounds from the medium, leak internal components and/or suffer osmotic stress. If such cells are preserved for further research or other uses, some caution must be exercised. In this respect, it has been published that low average power fs pulses are more than capable of porating cells, which totally preserved their viability but, nevertheless, made them to incorporate molecules from their surroundings (e.g., propidium iodide) [115]. The authors hypothesize that thermoelastic pulses are the source of this mild poration. This fits models of pressure stress and micro/nanocavitation in aqueous liquids by low-energy fs pulses [101,102], as well as recent models in which large pulsed thermal gradients (~10^9^ Km^−1^) across biological membranes lead to efficient poration due to fast (ns) increase in membrane fluidity [116].

Direct molecular thermal damage and pressure transients, driven by thermal processes, seem the most obvious sources of cellular alterations as a consequence of photothermal events. However, it is worthy to briefly discuss other heat-driven phenomena that are often overlooked, and which can also modify/interfere with cellular functions or structures during NIR optical trapping, perhaps in subtler ways. The first is the phenomenon known as thermophoresis. Thermophoresis is the movement of atoms, molecules and other microscopic entities (including biomolecules and micron-sized objects) when they are subjected to a temperature gradient [117]. Under this gradient, non-equilibrium thermodynamic conditions occur, to which molecules react by displacing. The direction and rate of such movement depends on the local electric fields that arise as a consequence of the thermal gradient. These electric fields appear because the solvation shells and diffusion rates of different chemical compounds change in different quantities under the same temperature difference [118]. All of this leads to differential diffusion and molecular flows. Within and in the surroundings of an optical trap important temperature gradients can take place, even if the absolute temperature change is small. For example, a temperature increase of 1 ºC would be considered mild. However, if confined to a 1 μm spot, it can set a temperature gradient of 10^6^ K m^−1^ with its immediate surroundings [119]. Thermophoretic flows have been successfully created within a living HeLa cell under localized subcellular gradients of 3–5 K [120]. Therefore, these seemingly innocuous temperature increases of a few degrees can interfere with the internal movement of cell constituents and, in consequence, lead to negative effects on cellular metabolism. A possible, although unlikely, example of this is provided by Ayano et al., who tried to explain the observed negative effects of optical trapping on *E. coli* appealing to an optical restrain at the molecular level, which would slow down or stop cellular metabolism [75]. As they employed a 1064 nm laser, it is possible that some kind of thermophoretic flow was imposed on the bacterial cells, interfering with biochemical reactions. However, the results reported can be perfectly explained by the direct optical excitation of ^1^O_2_, as discussed in Section 2.2 above.

Additionally, these local thermal gradients, for reasons very similar to the just mentioned thermophoresis phenomenon, can induce electric charge accumulation on microscopic particles. For a 30 K increase in a ~1–2 μm spot, fields of the order of 10^4^ Vm^−1^ have been calculated just besides the spot; and still 1–10 Vm^−1^ at 100 μm [121]. Even the simple heating of an aqueous NaCl solution, without colloids, has been calculated to induce a field of 200 Vm^−1^ locally for a ~1–2 μm spot and ∆T = 10 K [122]. The bottom line is that electric fields of a non-negligible magnitude can be induced at the micron scale for ∆T>1 K. These fields can have an impact on cellular processes, particularly if the biological sample is kept for a long time in the trap, on the doubtful assumption that a small temperature increase is not harmful.

Recently, a linked phenomenon was reported in the literature. Under a sustained steady thermal gradient imposed on a solution, solutes tend to concentrate according to their characteristics (mass, charge, solvation energies, etc.). If one or some of these molecules have acid-base activity, then a pH gradient is also established as a consequence of the heat flow in the system. In a phosphate buffered solution a difference of 2 pH units has been generated experimentally for thermal gradients of 8–30 K mm^−1^ for 1–4 h [123]. These are much weaker than the gradients in optical traps. Given the pH buffered nature of biologically-compatible fluids employed in research, it can be interesting to study if some kind of pH gradient is established during optical trapping, and its impact in biological samples. Indeed, this could offer an explanation for the results reported by Rasmussen et al. on pH changes in trapped bacteria [77].

Another thermal phenomenon, perhaps rarely found in optical traps except for fs pulses-driven ones, is water thermopolarization. This occurs when water is subjected to a sudden, fast (μs or less), quite localized (~microns) increase in temperature. A far from equilibrium situation is produced and the water molecules react to the thermal gradient by reorienting themselves in the direction of the heat flow. As water molecules are dipoles, this orientation induces a local polarization and it engenders an electric field [124]. For temperature gradients of 10^5^–10^8^ Km^−1^ transient electric fields of magnitudes 10^3^–10^6^ Vm^−1^ have been calculated [124,125]. Under pulsed laser optical trapping, as mentioned, most probable with fs pulses, the conditions can be the right ones for this kind of water thermopolarization to arise. Tentative calculated fields for typical optical trap parameters show magnitudes of 10–100 Vm^−1^ [126]. Although it is doubtful that thermopolarization has any real impact in optical traps, there are reports on transient electric signals after pulsed IR laser excitation of water samples [127,128,129]. Voltages in the order of 0.1–1 V were measured for centimeter-size water samples, matching values with fields of 10–100 Vm^−1^. Recently, modulation of several cellular features (Ca^2+^ dynamics, morphology, cytoskeleton) has been accomplished by means of a micro-heater based on 1064 nm laser heating of a micron-sized metal device [130]. The temperature gradients measured were of the order of 1–10 K μm^−1^ (10^6^–10^7^ Km^−1^) to be compared with the values just discussed above in relation to these thermal phenomena. To summarize, some important photothermal references are presented in Table 4.

In conclusion, it seems important to evaluate better the biological responses, particularly long-term ones, to heating in and around optical traps. It is undeniable, in light of the many reports, that immediate, lethal damage is not inflicted under most situations. However, more subtle interferences can be occurring, which can lead to misleading conclusions.

## 4. Mechanical Damage in Optical Tweezers

The most obvious sources of biological stress and damage in optical tweezers are photochemistry (Section 2) and photothermal processes (Section 3). A frequently overlooked phenomenon, however, is the excitation of mechanical activity in the sample by the optical trap. It arises because of the coupling between heat produced by photon absorption and the temperature-dependent changes of several material properties (e.g., thermal expansion, convection, induced flows, etc.) [100]. This mechanical activity can be understood at several levels. First, at the whole cell level, where shaking and translation can take place. Second, at the subcellular/organelle level, where internal vibrational modes can be excited. Third, at the supramolecular level, for example, with mechanical entrainment of the cellular membrane. Finally, at the molecular level, where conformational alterations can occur with biological implications. Examples of these mechanical effects will be provided in what follows, to warn the reader but also to spur deeper research into this relatively unexplored area.

A decade ago, Zhang and Liu already advised about the necessity to better assess opto-mechanical coupling in optical traps [11]. Here, these mechanical effects are not those related to gross thermal phase changes like boiling or cavitation. Those effects have much more to do with the results reported in Section 3, although their explosive nature immediately warns the researcher about their occurrence. The average powers involved to induce cavitation, for example, frequently are quite above the usual ones employed for optical trapping. The mechanical effects described below are more subtle in their nature and, therefore, can occur without immediate noticing by the manipulator/observer. They are the result of coupling between an initial thermal event (photothermal effect) and the mechanical properties (i.e., thermal expansion) of the trapped sample. The ensuing thermoelastic waves can interact and alter cells and biological compounds in different ways.

At the whole cell level the thermal action of the optical trap can modify the temperature and viscosity of the surrounding solvent. This can alter the equilibrium of forces and produce undesired movements and displacements of the sample. As already mentioned, changing the numerical aperture or the focal distance close to a solid surface can change the temperature increase in the trap, everything else kept equal [109]. One consequence is that undesired flows can be established around the sample. Recently, it has been published that low-energy NIR fs pulses at 50 MHz can induce noticeable changes in the trap microviscosity [131]. These changes translated to kHz oscillations in the trap´s stiffness, therefore acting like a microscopic “spring”. Should a cell be located at this trap, it would be subjected to kHz vibrations.

The cell, as a unit, and subcellular structures (nucleus, Golgi, etc.) can undergo mechanical oscillations as a result of being exposed to thermomechanical waves resulting from thermal expansion/contraction cycles. For example, Ng et al. published a very interesting paper in which leukemia cells were irradiated with a CW 1064 nm laser but in a scanning mode [132]. In this way, cycles of thermal expansion and contraction were induced in the cells. The remarkable point is that almost complete cell death was obtained for certain scanning frequencies (see Figure 20). Non-scanning laser exposure with the same total dose induced no damage. There is no simple trend in the frequency-cell death plot. Instead, it seems that certain frequencies induce a resonating action that severely affects cells. Further research by this group reinforces the phenomenon, with a working model that relates the mechanical resonances to the disruption of the cortical cytoskeleton-nuclear envelope and consequent lethal damage [133]. In this respect, other recent publications point to the particularly relevant role of the nucleus as a vulnerable spot for pulsed thermal treatments [134,135]. With a temperature rate of change of 1–20 Ks^−1^ the nucleus undergoes reversible expansion/contraction (depending on the ionic properties of the medium). This can set intracellular vibrations that can have long-term effects on the cell [135]. In fact, nuclear-cellular mechanotransduction is a current hot research topic, as it seems the nucleus is directly sensitive to pure mechanical cues transmitted by the cytoskeleton and transduced by conformational molecular changes (e.g., chromatin compaction) and pressure-sensitive proteins (ionic channels) [136,137].

Vibrations of 10–1000 Hz but lower amplitude have been shown to increase the uptake of molecules into the cell, with gene delivery proven feasible [138]. Again, only certain frequencies, very similar to those already mentioned before and displayed in Figure 20a, are efficient at increasing cell permeability. Experiments carried out with an optically-trapped polystyrene bead show that this bead can be employed as a sort of subcellular-sized “*jackhammer*”, promoting membrane indentations and intracellular Ca^2+^ transients, all with forces in the pN range [139]. This experiment points in the direction that mechanical waves can, indeed, drastically modify the intracellular medium. The trapped bead “*drilling*” frequency is indicated as 1 Hz. In [138] mechanical shaking at 10-800 Hz was effective at increasing molecular and nanoparticle uptake, with a frequency of 100 Hz being particularly effective towards this goal. This is to be connected with the ~kHz oscillations in trap´s stiffness reported by Mondal et al. with their fs pulsed trap [131].

Moving further down in scale, mechanical waves can directly interact with biological membranes. For example, in order to explain the puzzling effects of ultrasounds and microwaves in neurons, different working models and experimental results have been proposed and reported in the last decade. Shneider et al. advance a model in which ultrasound/microwaves induce standing or travelling mechanical waves in the membrane [140]. These waves can alter the density distribution of membrane-bound ionic channels, leading to enhancement or blocking of the electric signal. Another group provides two mechanisms, not mutually exclusive, in which thermal transients engendered by the ultrasound/microwave lead to thinning of the membrane with a concomitant change in capacitance, driving inductance currents [141]; or favor intramembrane nanometric cavitation which also provokes currents [142]. The connecting point of these papers and models with optical tweezers is that the thermal transients that lead to these changes are very weak, functioning with a negligible increase in temperature (<<1 °C). Therefore, very low intensity (photo)thermal processes taking place in the optical trap, classically considered as insignificant, can, nevertheless, have a substantial impact in the cellular physiology if this kind of thermomechanical events are happening.

Finally, thermal modulation of molecular conformations can occur in optical traps. For example, trapping of T4 virus DNA with a CW 1064 nm laser in aqueous solutions led to cyclical transitions from the expanded coil state to the condensed compact state of the macromolecule [143,144]. No cycling was observed in D_2_O, which does not absorb NIR light at 1064 nm, thus favoring a thermal mechanism of action. The authors conclude that other polymers can undergo similar thermally-driven transitions under the adequate circumstances. These processes can take place within living cells and, more importantly, in solutions where assessment of other features (e.g., mechanical properties) is evaluated while the molecule of interest is optically trapped. In consequence, it is advisable to take in account the possibility that the biological sample can suffer conformational changes because of the perturbing action of the optical tweezers.

## 5. Strategies to Avoid Damage in Optical Tweezers

Once different kinds of damage and sample perturbation in optical tweezers have been introduced and discussed, it seems quite relevant to provide some measures and action lines to avoid or, at least, minimize any damage that a biological sample may suffer as it is exposed to an optical tweezers. It is fair to say, nevertheless, that a perfect strategy does not exist, and that the researcher must find a practical compromise depending on the different parameters of the particular experiment.

By now, it should be clear that NIR wavelengths are, by far, the most successful in dealing with biological samples for optical trapping. UVA wavelengths imply a direct photochemical damage even at the lowest doses. Visible wavelengths also lead to fast one-photon photodamage, particularly below 600 nm [42,44,47,49,50]. In the IR region wavelengths longer than 1100 nm induce important vibrational excitation of water and heat evolution. Thus, the range 750–1064 nm has been the most employed to implement optical traps. Within this spectral range, though, care must be taken as there are some molecular oxygen absorption bands that can promote excitation of singlet oxygen, with deleterious biological and biochemical effects (see Section 2.2) [62,71,73,87,92]. From several experiments of viability preservation (measured in different biological models and with different viability markers), it can be concluded that wavelengths in the ranges 810–860 nm and 940–960 nm are the least damaging.

The spatial patterning of the light in the trap can also help to reduce any damage by reducing the light dose to which the sample is exposed to. Holographic traps in which light intensity is reduced and several or many objects can be simultaneously trapped, is an option to consider [145]. Very recently, Liu et al. have described a modification of the microscope objective of the trap to obtain a “*light funnel*”, a dark cone with “*walls*” made of light [146]. This funnel can confine and trap objects of micrometric size without exposing them to the high light fluences typical of optical traps. Another strategy to reduce the photon load has been advanced by Koss et al. and consists in trapping inert microscopic beads (e.g., silica) in groups and then employing them as “*grips*” or “handles” to manipulate the biological target [147]. However, it adds complexity in the manipulation as direct control over the sample is lost and it can move away from the “*handle*”. Mixed photonic-material approaches, that reduce the light dose, have been proposed during the last decade too. Traps based on photonic crystals [18,29] or plasmonic tweezers [145] employ lower light intensities and provide flexibility to rearrange the trapping pattern. However, they are 2-D tools, dependent on a surface that becomes the trapping agent when it interacts with the light. Moving to a 3-D patterning is a challenge currently being faced by these traps. The same can be said at the moment of another approach, the optoelectronic tweezers based on light-excited ferroelectric materials, which works efficiently in 2-D but still has a long road to attain 3-D control [148].

Another front where damaging action can be controlled is the temporal features of the optical trap. Basically, the researcher faces two main options: CW or pulsed laser light. A CW laser excitation offers a simpler system and trapping efficiency seems a little higher than with fs pulses [99,103]. However, NIR fs pulsed traps offer trapping and non-linear excitation at the same time, which can be an asset to consider. With the optical energies and powers typically employed in fs optical tweezers, second-order non-linear processes seem greatly favored over third-order ones [149]. Therefore, the possibility to excite molecules at half the trapping wavelength must be considered, in particular if the experiment involves exogenous fluorophores. Traps based on ns pulsed lasers are discouraged, as they promote a large variety on undesirable photochemical, plasma and thermal issues [44,70,76]. An alternative to real pulsed laser systems is to arrange the optical trap with a CW laser in a scanning/pulsed mode. In this case there are no optical peaks in power leading to undesirable non-linear or thermal effects. The CW beam can be chopped or turned on/off at a certain frequency to set up the trap. Less photodamage was reported with this approach by Mirsaidov et al. as compared with full time excitation [92]. In fact, reducing the excitation frequency of a fs pulsed system from the usual 80 MHz to 200 kHz and, additionally, decreasing the focusing angle (wide-field illumination) greatly increases the performance of an optical trap as recently reported [150].

A critical parameter for optical tweezers manipulating biological samples and organic molecules is the presence and activation of molecular oxygen. Plenty of information has been provided on this particular issue in Section 2.2. In order to keep oxidative damage in check, several strategies can be undertaken. In the first place, the O_2_ absorption bands in the NIR should, obviously, be avoided (see Figure 5) [62]. In this sense, it must be remarked again that optical traps relying on 1064 nm excitation will be producing ^1^O_2_ in the trap medium. Quenchers added to the medium can help in preventing or delaying oxidative damage. For example, sodium azide (NaN_3_) is a very efficient ^1^O_2_ quencher and has been employed in different experiments [62,67,76]. However, it blocks the mitochondrial electron transport chain; therefore it is toxic for cells in the long-term and must be used with care. DABCO (1,4-diazabicyclo[2.2.2]octane) is another efficient ^1^O_2_ quencher, more tolerable by living cells. β-carotenes or α-tocopherol are organic compounds with a proven efficiency to decrease ^1^O_2_ oxidative action [66,88,151,152]. Sacrificial reducers (i.e., antioxidants), like ascorbate, can be employed but they engage in chemical reactions with ^1^O_2_, thus other ROS (e.g., superoxide or H_2_O_2_) and radicals will be produced close to the optical trap.

A particularly interesting approach to the oxygen problem is to eliminate it altogether, bypassing the need to deal with ^1^O_2_ or other ROS. This is achieved by employing biochemical scavenging systems, based on oxygenases/oxidases. For example, a mixture of glucose oxidase plus catalase (GOC) efficiently consumes O_2_ and reduces pO_2_ in the trap. This approach was employed by Landry et al. to minimize oxidative damage on DNA in their tensile measurement experiments [79]. Also, Min et al. found an important decrease in bacterial damage with a 1064 nm trap in the presence of GOC [153]. To be fair, Neuman et al. already considered the possibility to employ both oxygen scavengers and ROS quenchers in order to avoid photodamage in optical traps [72]. These systems are not without drawbacks, however. It is obvious that consuming O_2_ in the medium can have a very negative impact on living organisms, particularly aerobic ones. Additionally, the GOC system (and others) releases organic acids as a side product of the oxygen-consuming processes. This lowers the medium pH which, again, can have negative biochemical or biological effects in the long-term. Alternatives have been proposed, such as the pyranose oxidase-catalase, which releases an aromatic ketone that does not alter the pH [154]. This is an interesting possibility to avoid both oxidative and pH-related damage.

A special comment deserves the use of deuterium oxide (D_2_O or heavy water) in optical traps. Some researchers have employed D_2_O to minimize the thermal load in 1064 nm optical traps [85,143]. This is due to the very low absorption of 1064 nm photons by heavy water. From the point of view of avoiding an increase in temperature, this strategy is perfect. However, the lifetime of ^1^O_2_ in D_2_O is ~20 longer than in H_2_O. Therefore, any oxidative damage will be exacerbated in conditions of optical trapping in D_2_O or mixtures of D_2_O and H_2_O (particularly at 1064 nm). Also, for long incubation times (hours) D_2_O has been shown to be cytotoxic [155].

Finally, some comments in regards to methodologies that can reduce the thermal load in optical traps will be presented. Wavelengths between 700 nm and 1100 nm produce the lowest temperature increases (see Section 3). Removing wavelengths exciting water vibrations (see Figure 3) and oxygen (see Figure 5) leaves ample spectral regions in the 700–1000 nm where minimal thermal stress should be expected [110]. Reducing the photon flux would, necessarily, decrease heat production. Therefore, optical traps based on scanning, “*hollow*” light structures, and/or pulsed lasers should, in principle, heat less than a continuous light excitation [99,145,146]. Reducing the pulse frequency in a fs laser from 80 MHz to 200 kHz has been shown to be quite effective in reducing the temperature increase by one order of magnitude (from ΔT = +5 °C to + 0.25 °C) [150]. Recently, “*pulsing*” of CW NIR lasers (808 nm vs. 980 nm) was an efficient strategy to avoid trap heating, particularly at 980 nm which vibrationally excites water molecules [156]. In CW mode, the 980 nm laser produced a ΔT = +21 °C at 130 mW. Modulating the emission decreased ΔT to +15 °C (30 ms pulse width), +10 ºC (20 ms pulse width), or +1 °C (10 ms pulse width). This “*pulsing*” approach indeed reduces the thermal mode but thermomechanical processes are more readily induced (see Section 4). Then, caution is advised to check for possible thermoacoustic effects on the sample, especially at 100–1000 Hz frequencies [138].

Micro-flows have also been considered to reduce the temperature increase in optical traps. Facilitating micro-flow evolution or thermal conduction to nearby structures by adjusting the spatial location of the trap in the working chamber helps in improving these thermal aspects [109]. Others have coupled a heating/cooling cap to the microscope objective to induce convective currents that should help in dissipating heat [157]. These authors also advanced another strategy based on a double beam trap, in which a non-heating laser (830 nm) pumps the optical trap and another laser (975 nm) induces a controlled temperature rise which, in turn, drives convective currents promoting overall cooling the trap. Creating optical traps within microfluidic circuits in which micro-flows carry away the heat is another successful strategy [158]. An interesting option is to adapt the medium composition to better deal with these issues. For example, Chowdhury et al. recently described the catastrophic effect of intense 1064 nm trapping on red blood cells (see Figure 19) [113]. However, changing the cell medium composition from ionic (phosphate-buffered saline, PBS) to non-ionic (sucrose), while maintaining isotonic conditions, resulted in an increase in the threshold for catastrophic thermal collapse from 350 mW to 600 mW. The authors relate this to changes in the thermal conductivity linked to the ionic/non-ionic nature of the solutes.

In conclusion, several parameters can be modified to decrease either heat evolution in the trap or heat dissipation from the trap. Depending on the particular necessities of each particular experiment, some or all can be implemented. Nowadays, cheap and reliable diode lasers emitting in the 800–900 nm range are available, which should translate in research teams moving from classic 1064 nm traps to this near-visible region. Some discussed strategies to reduce damage in optical traps are compiled in Table 5.

## 6. Outlook and Perspectives

To finish this review, some ideas and new angles of view for the topics previously discussed will be provided. It is more than clear that optical tweezers have revolutionized many fields, nonetheless biotechnology and biophysics. Still, there are areas in which there is plenty of room for improvement, one of them the reduction or elimination of damage that comes from the optical trap itself. Better optics to improve trapping and reduce light intensity on the sample are under development, as introduced in the recent and relevant review by Bunea and Glückstad [145]. Bessel, Airy or Laguerre–Gaussian beams are being studied for optical trapping, for the advantages they offer over conventional optics.

In many instances it can be desirable to increase the optical interaction with the biological sample, in order to alter it in some desired way. Such is the principle of the optical tweezers-scissors or tweezers-scalpel. The optical trap can hold the sample and, at the same time, carry out some kind of manipulation (subcellular transport, membrane poration, organelle ablation, etc.) [12,13,105,145]. A particular example was presented above in Figure 16b, in which an organelle was extracted from a yeast cell by employing this tweezers-scalpel approach [105]. Pulsed fs lasers seem to provide better spatial and thermal control for subtle “*light scalpel*” procedures [159]. A dual beam setup, in which one wavelength creates the trap and another is employed to alter the sample, is an emerging trend although it increases the methodological complexity [157].

The possibility to influence the migration and growth direction/rate of cultured cells is another poorly explored area. Recently, a series of photothermal microirradiation experiments have concluded that it is quite feasible to modify the growth of cells or their migration in certain selected directions just by a laser-driven temperature increase [160,161,162,163]. In fact, it has been shown that yeast budding and colony growth can be directed along a linear optical trap [164]. The authors propose a mild thermal excitation to explain the observed cell response but other alternatives, such as low ROS production stimulating the yeasts, cannot be disregarded particularly because they employed a 1064 nm laser. For additional sources on the microscopic mechanisms and applications of photothermal manipulation the reader is directed to the recent review [165] and references therein.

A very promising research area is that of parallel optical trapping-membrane fusion. Cells [166] and other biological membrane-containing structures (e.g., vesicles, liposomes, etc.) [167] can be fused when brought under very close contact and then inducing a quick temperature change. A single-beam or dual-beam optical tweezers could hold structures of interest in very precise spatial positions or arrangements, and then proceed to induce a fast photothermal pulse in order to obtain merged structures. Alternatively, suitable nanoparticles (carbon-based, gold) can be employed to better localize the point of fusion (nanometric) and then act as photothermal agents (“*rivets*”) on the spot.

Optical traps offer an overlooked approach to study redox biology in trapped cells. Practically all efforts in the optical tweezers literature have been directed to avoid ROS generation in this sense. However, here exists a unique environment to precisely produce ROS in a controlled way in a selected cell or group of cells. For example, trapping for short times with a 765 nm beam can be employed towards such goal. Or a dual beam for trapping/producing ROS (e.g., 830 nm + 765 nm). Currently, it is accepted that ROS have a critical signaling role in many cellular processes and responses [63,65,168]. In recent years, cell microirradiation and direct optical excitation of ^1^O_2_ with a 765 nm laser has provided very interesting results in the field of redox biology [62,66,88,151,152]. For example, everything else equal (laser power, light dose, wavelength, etc.) microirradiation of the cell in the cytoplasm or the nucleus brought about radically opposite proliferative responses: irradiation in the cytoplasm enhanced proliferation while in the nucleus it was delayed [169]. Parallel experiments with detached cells are lacking and could provide a great deal of information to be compared with results in adherent cells. Optical traps will be an invaluable tool to achieve precisely this. Another approach could be to trap cells, previously incubated with a photosensitizer, at low power with a NIR laser. Then, for short times, increase the laser output to maintain the trap and induce a two-photon excitation of the compound, which would lead to physiologically meaningful pulses of ROS [170]. This could expand already published results on redox modulation of cell responses with photosensitizers [171,172].

Finally, optical tweezers offer a unique opportunity to have a precise control over the studied object at the micro-nanoscopic level and, at the same time, act as a probe to assess information at different levels (composition, structural, conformational, etc.). The study of the mechanical properties of different molecules has been possible to a great extent thanks to optical tweezers [10,22,31]. Temperature can be suddenly increased in the optical trap, allowing temperature-jump experiments for molecules [173]. This temperature-jump methodology should be exported to single cell studies to evaluate potentially relevant responses. Recently, detailed molecular analysis of cells and subcellular components has been obtained taking advantage of optical tweezers that provide Raman spectroscopy analysis at the same time [174,175]. Thus, this methodology offers the adequate setup to trap and interrogate the biological sample at the same time.

These are just some of the many new applications and methodologies, currently under development or foreseeable, that optical tweezers have to offer in the biological and biophysical arenas. Undesirable and negative effects of optical traps provide, nevertheless, new methodological opportunities and novel paradigms when considered from different angles. It has been the goal of this review to present the main damage mechanisms taking place in optical traps for trapped biological samples, while providing means and strategies to avoid/minimize their impact. Also, to provide some out-of-the-box suggestions to actually take advantage of those mechanisms in order to obtain more biological information and further advance in the fields of biology and biophysics.

## Figures and Tables

**Figure 1 micromachines-10-00507-f001:**
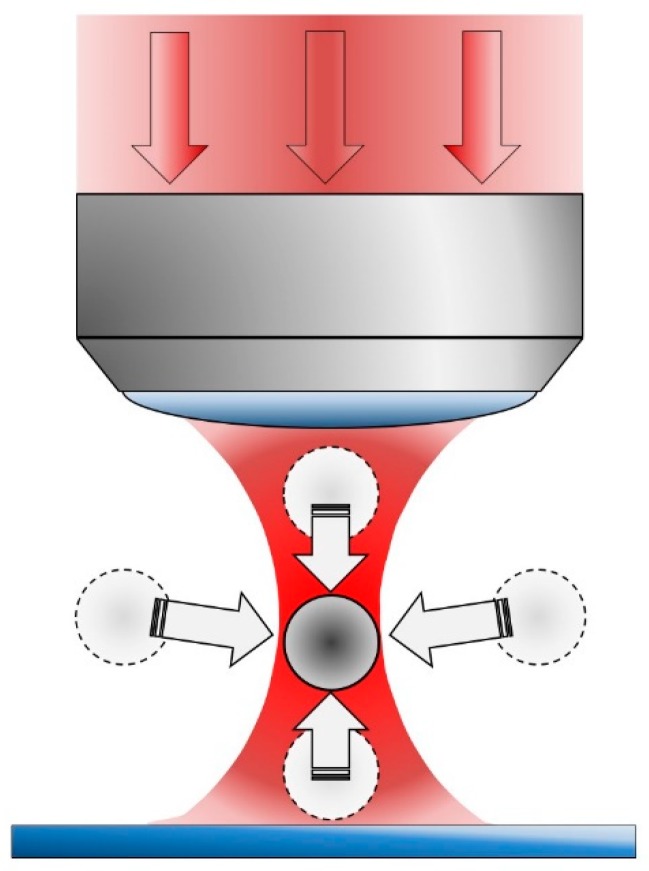
Simplified depiction of optical tweezers. The incoming laser beam (red arrows) enters the objective from the top and focuses at a certain point determined by the magnification factor and numerical aperture of the system. Optical forces create a potential well at the focal point (beam waist) where a microparticle (grey sphere) with a higher refractive index than the surrounding medium can be trapped. If the original sphere´s location does not coincide with the optical trap (blurred spheres), the optical forces are not under equilibrium and a net force displaces the particle towards the trap (grey arrows).

**Figure 2 micromachines-10-00507-f002:**
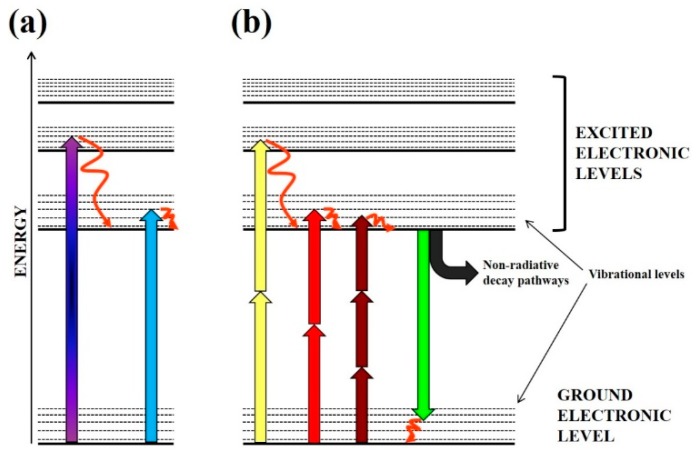
Jablonski diagrams showing simplified photon absorption. (**a**) Linear optical absorption in which a single photon pumps a molecule to an excited state. In this scheme, a blue photon provides enough energy to reach the first excited state and an ultraviolet (UV) photon pumps the molecule to the second excited state. Electro-vibrational (vibronic) deexcitation then ensues, releasing heat (wiggly red arrows); (**b**) non-linear optical excitation, in which two yellow photons add together to pump the second excited molecular state (equivalent to one UV photon). Alternatively, two red or three near-infrared (NIR) photons provide enough energy to pump the first excited state. Heat can also evolve in these non-linear processes. From the first excited state either luminescence (fluorescence or phosphorescence; green photon) or non-radiative deactivation (e.g., photochemistry) can occur.

**Figure 3 micromachines-10-00507-f003:**
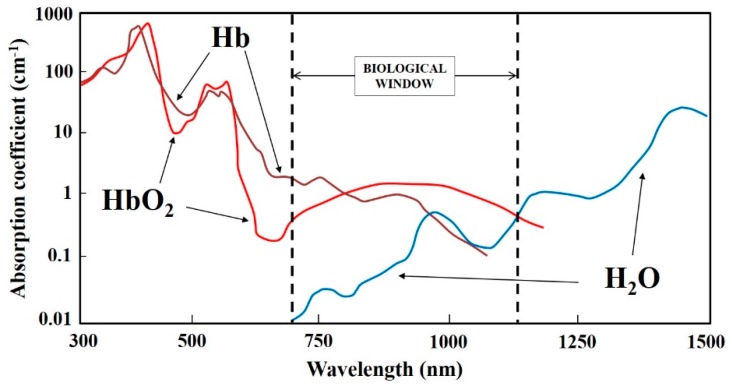
Optical absorption of hemoglobin (Hb), O_2_-binding hemoglobin (HbO_2_) and water (H_2_O), from the ultraviolet A (UVA) to the NIR. Hb and HbO_2_ absorption curves here can loosely serve as examples of the absorption of other biomolecules (cytochromes, etc.) in the cell. The so-called biological window encompasses a spectral region broadly bounded between 700 nm and 1100 nm. Within these boundaries optical absorption is minimal, although not negligible, and thus represents the optimum for wavelengths to be employed in optical tweezers.

**Figure 4 micromachines-10-00507-f004:**
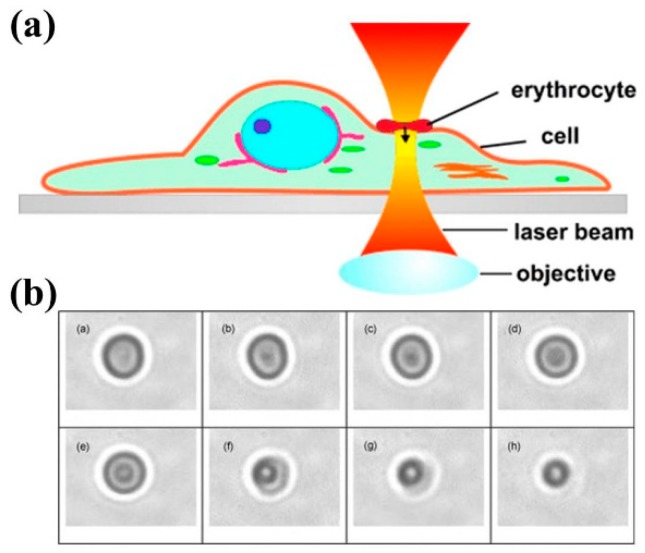
(**a**) Red blood cells (RBC) handle-optical tweezers scheme. A chemically-fixed erythrocyte is first attached to the surface of the target cell. Then, by optically-trapping the erythrocyte, it is possible to indirectly deform or alter the target cell. (Reprinted with permission from Wiley InterScience [45].); (**b**) RBC collapse sequence during optical trapping with a 785 nm laser at 9 mW. Note the important cell shrinking with trapping time and the cell mass condensation at the trap focus. Trapping times, (a) to (h), are: 22, 25, 28, 31, 33, 33.6, 33.9 and 36 seconds. Reprinted with permission from [52].

**Figure 5 micromachines-10-00507-f005:**
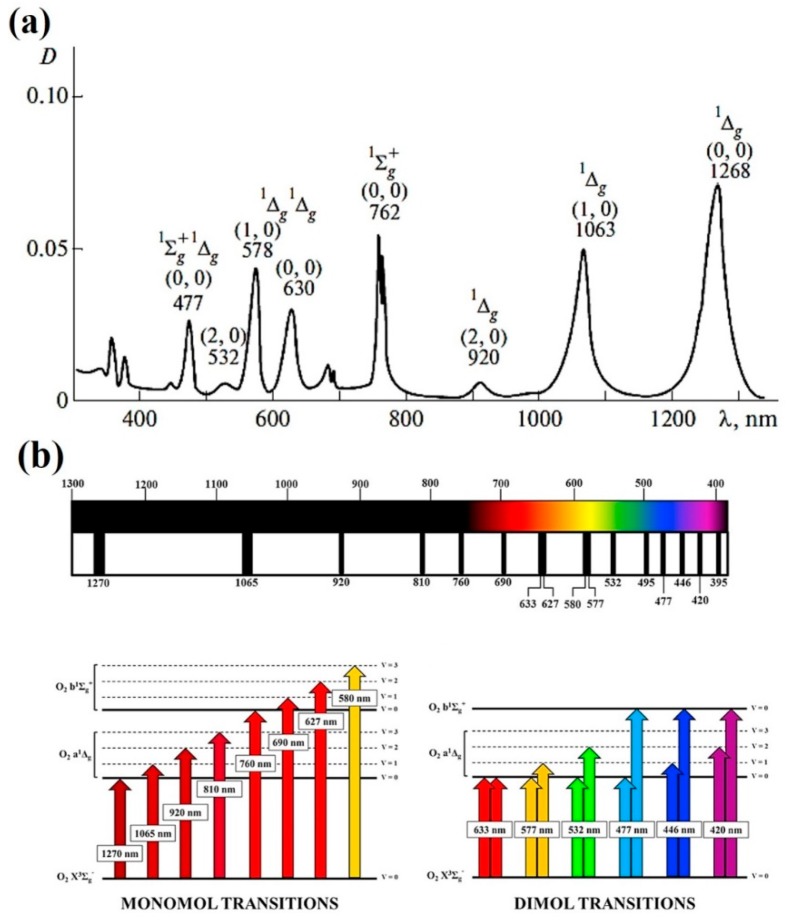
(**a**) Optical absorption spectrum of high-pressure molecular oxygen in the region 300–1350 nm. The symbols represent the final singlet oxygen state reached, and the numbers in parentheses the final and initial vibrational levels of the particular transition. (Reprinted with permission from Springer Nature [55].); (**b**) scheme showing the optical transitions introduced in (**a**) among the different electronic and vibrational energy levels. Note that both monomol and dimol (see main text) transitions are possible. Reprinted with permission from [62].

**Figure 6 micromachines-10-00507-f006:**
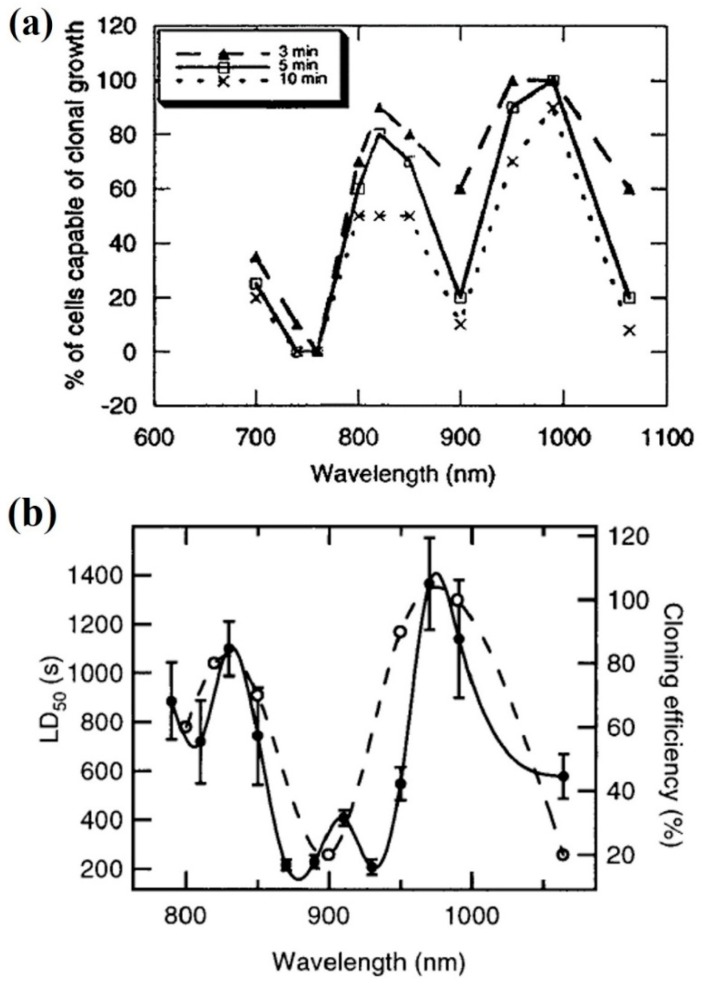
(**a**) Damage action spectrum on Chinese hamster ovary (CHO) cell cloning efficiency after trapping with a tunable lasers between 700 and 1064 nm. Trapping times were 3, 5 and 10 min (see inset box). Notice significant cloning decreases at 740–760 nm, 900 nm, and 1064 nm. (Reprinted with permission from Elsevier [71].); (**b**) Action spectrum for bacterial inactivation (left axis, black symbols, solid line). Fast inactivation occurs at 870 and 930 nm, and also at 1064 nm with lesser severity. Data from [71] has been plotted for comparison (right axis, open symbols, dashed line). Fit between both action spectra is remarkable. Reprinted with permission from [72].

**Figure 7 micromachines-10-00507-f007:**
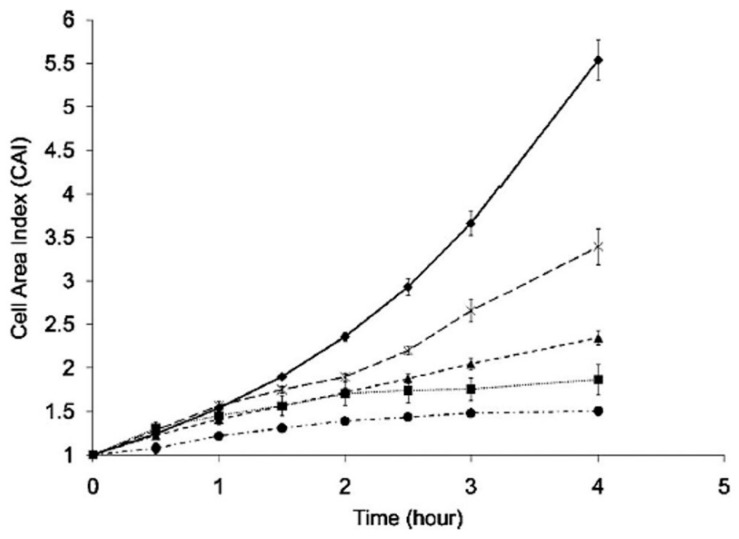
Effect of CW 1070 nm laser trapping on yeast growth (measured as Cell Area Index) for very long times (up to 4 h). Very low powers were employed (from top: control, 0.7 mW, 1.3 mW, 2 mW, and 2.6 mW). All powers produced a growth delay, the more significant the larger the power. A certain degree of adaptation can be observed for 0.7 mW and, perhaps, for 1.3 mW (note change in slope). Reprinted with permission from [80].

**Figure 8 micromachines-10-00507-f008:**
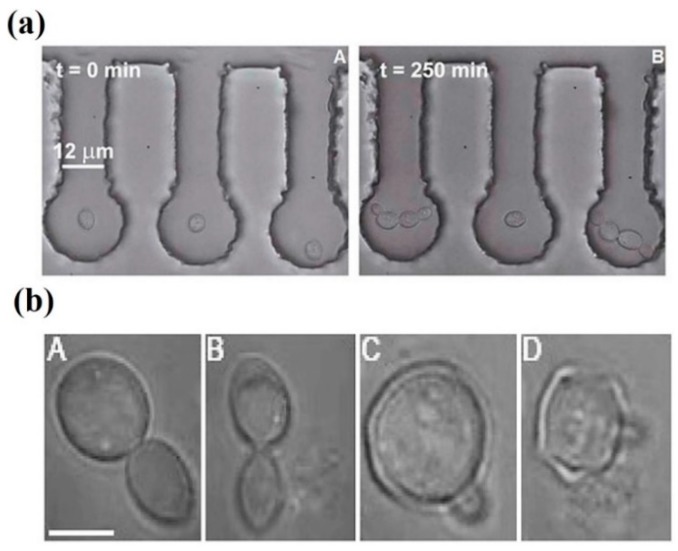
(**a**) Dynamics of division of *Saccharomyces cerevisiae* cells stressed by optical tweezers in a microfluidic chip. The cell in the middle micro-chamber was optically trapped with laser power high enough to halt the division, while the two peripheral control cells continue budding.; (**b**) Images of two different *S. cerevisiae* cells before (A,C) and after (B,D) cell-wall rupture caused by 15 min of optical trapping at the wavelength 1064 nm with trapping laser power 76 mW. The rupture always occurred during the new bud formation, more than 60 min after the end of optical trapping. Scale bar: 5 μm. Reprinted with permission from [85].

**Figure 9 micromachines-10-00507-f009:**
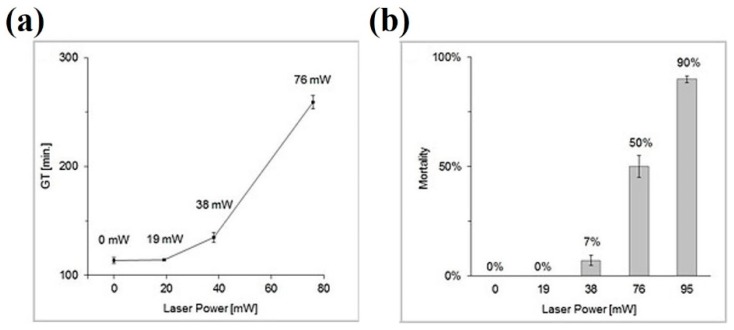
(**a**) Generation time (GT) of *S. cerevisiae* cells trapped for 15 min in optical tweezers with laser wavelength 1064 nm and trapping laser power in the range from 19 to 95 mW. The 0 mW point corresponds to the control unexposed cells. (**b**) Mortality (M) of optically trapped *S. cerevisiae* cells under experimental conditions identical to those in the GT plot. On average, 13 samples were used for each data point. Error bars correspond to the standard error of the mean (SEM). Reprinted with permission from [85].

**Figure 10 micromachines-10-00507-f010:**
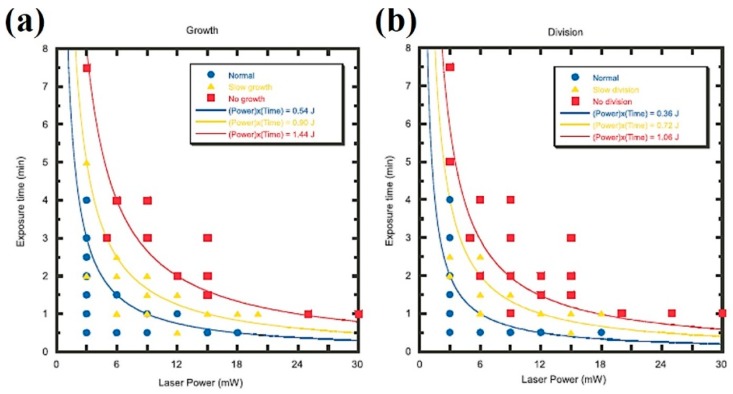
Damage to growth and division capabilities of *E. coli* under various 1064 nm laser power and trapping time conditions. The blue circles in the graph represent “normal” growth/division, whereas the black triangles and red squares show “slow” or “no” growth/division, respectively. (**a**) Damage estimated from changes in growth rate; (**b**) damage estimated from changes in division rate. Reprinted with permission from [75].

**Figure 11 micromachines-10-00507-f011:**
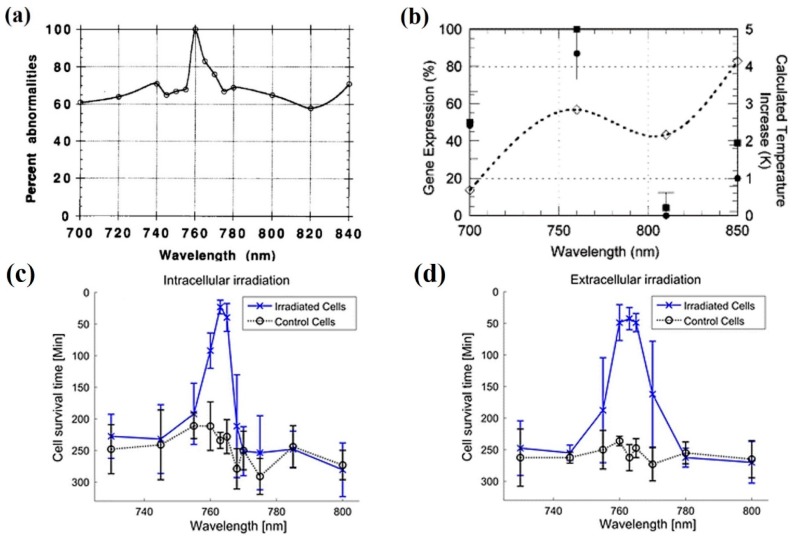
Action spectra for biological damage in the 700–850 nm region. (**a**) Action spectrum for chromosomal aberration induction in rat kangaroo cells. Note the peak in damage at 760 nm. Reprinted with permission from [86]; (**b**) Action spectrum for heat shock protein activation in transgenic *C. elegans* (left axis, solid symbols). A peak appears at 760 nm. On the right axis (open symbols, dashed line) the calculated temperature increase for each wavelength (see Section 3). Reprinted with permission from [87]; Necrosis induction action spectra for (**c**) intracellular or (**d**) extracellular irradiation of HeLa cells. There is a clear decrease in cell survival time in the region 760–765 nm. Reprinted with permission from [88].

**Figure 12 micromachines-10-00507-f012:**
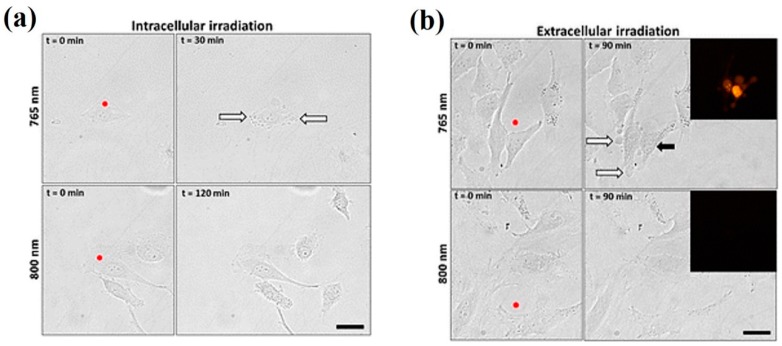
Bright-field images showing responses of HeLa cells to laser irradiation at 765 and 800 nm. The red spots indicate the irradiation site. (**a**) Cells exposed to intracellular irradiation for 10 min at 91.5 mW on-stage. Thirty minutes after the start of irradiation, the cell exposed to 765 nm light clearly shows signs of necrosis (e.g., membrane bubbling, white arrows). (**b**) Cells exposed to extracellular irradiation for 30 min at 91.5 mW on-stage. Ninety minutes after the start of irradiation, the cells exposed to 765 nm likewise show signs of necrosis (bubbling—white arrows-, pycnotic nucleus—black arrow). The insets show images of the same region based on the fluorescence of propidium iodide. In all cases, cell damage was not observed upon irradiation at 800 nm. Scale bar = 20 μm. Reprinted with permission from [88].

**Figure 13 micromachines-10-00507-f013:**
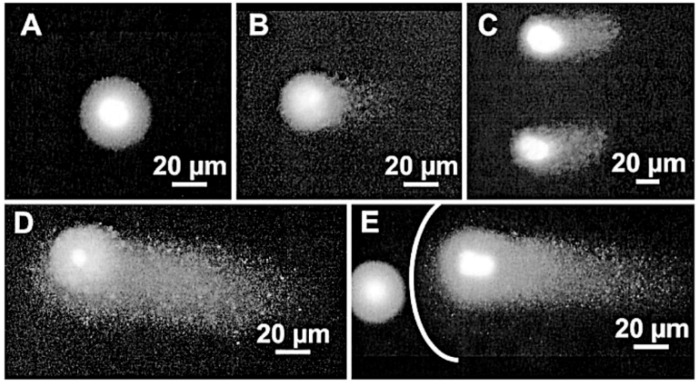
Images of NC37 lymphoblast cells after laser microirradiation and comet assay evaluation, showing different levels of DNA damage. Panel (**A**): Typical control cell that has not been exposed to laser irradiation. The typical round shape of an undamaged cell is maintained. This indicates that the procedure of agarose embedding and electrophoresis does not induce DNA damage by itself. Panel (**B**): A comet resulting after exposure of the cell to 800 nm at 60 mW for 60 s. A slight increase in the DNA damage was observed compared to the control cells. Panel (**C**): Two comets exposed to a Ti:sapphire laser at 760 nm at 60 mW for 60 s. Compared to the same exposure at 800 nm, the amount of damaged DNA is strongly increased. Panel (**D**): The DNA damage shown in panel (**C**) can be increased further at the same wavelength by increasing the laser power to 120 mW at the same exposure time. Panel (**E**): This image is taken from the border of the irradiation circle. The left cell has not been irradiated, whereas the right one was exposed to 240 mW for 60 s at 1064 nm. The resulting DNA damage is less than that obtained with the half energy dose at 760 nm; see panel (**D**). Reprinted with permission from [42].

**Figure 14 micromachines-10-00507-f014:**
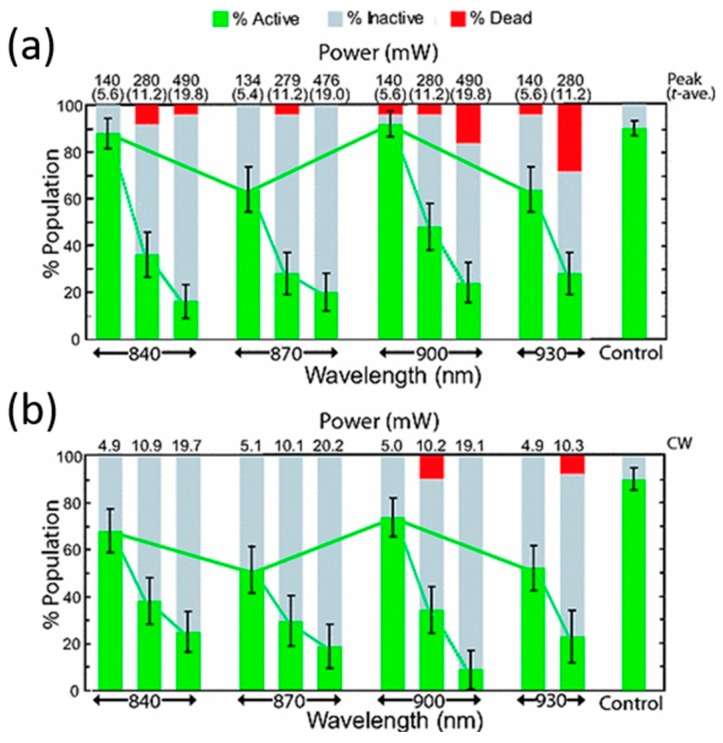
Viability as a function of NIR wavelength, power for time-shared and static optical traps. In panel (**a**) 5 × 5 2D arrays of *E. coli* bacteria incorporating the plasmid GFP-M1 (green fluorescent protein activated by isopropyl β-D-thiogalactopyranoside exposure) are assembled using a time-shared optical trap with the specified wavelength and power. In each case the cells in the microarray are held for about 8 min prior to gelling. The peak power is indicated along with the corresponding time-averaged power in parentheses. The bar graph represents viable, active bacteria (green), inactive bacteria (gray), and dead bacteria (red) for each wavelength and power. Viability decreases nearly linearly with increasing power, and peaks at λ = 840 and 900 nm. Panel (**b**) shows similar results to panel a) but instead using a CW beam to form the 5 × 5 2D arrays of *E. coli*. The static CW in optical traps ranges from about 5 to 20 mW at the specified wavelength. Again, in each case the cells in the microarray are held for about 8 min prior to gelling. The CW viability tracks that found for the time-shared trap at about the same time-averaged power. The right side of the corner shows control bacteria, non-trapped but encapsulated in the hydrogel spot. Reprinted with permission from [92].

**Figure 15 micromachines-10-00507-f015:**
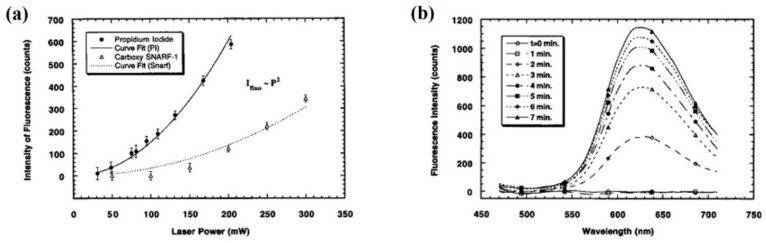
(**a**) Dependence of fluorescence intensity on pump (trapping) laser power for the fluorophores propidium iodide (PI) and Snarf. The intensities vary with nearly a square-law dependence on laser power for both dyes, a behavior consistent with a two-photon absorption process.; (**b**) time evolution of the fluorescence spectrum from a PI-labeled sperm cell. The increase in fluorescence intensity with increasing time provides evidence for the real-time monitoring of the physiological state of an optically trapped cell, in this case indicating the onset of cell death; (**c**) fluorescence emission spectra from the DNA of living human sperm cells stained with acridine orange. Cells trapped with a CW laser exhibit green fluorescence (525 nm) and show no evidence of structural changes or photodamage. When trapped with laser pulses, the emission exhibits a strong color change to the red (645 nm), indicating structural DNA damage.; (**d**) cloning efficiency versus exposure time for CHO cells for multimode beams and single-frequency CW beam (760 nm, 88 mW). (**a**–**c**) Reprinted with permission from [70]; (**d**) Reprinted with permission from [90].

**Figure 16 micromachines-10-00507-f016:**
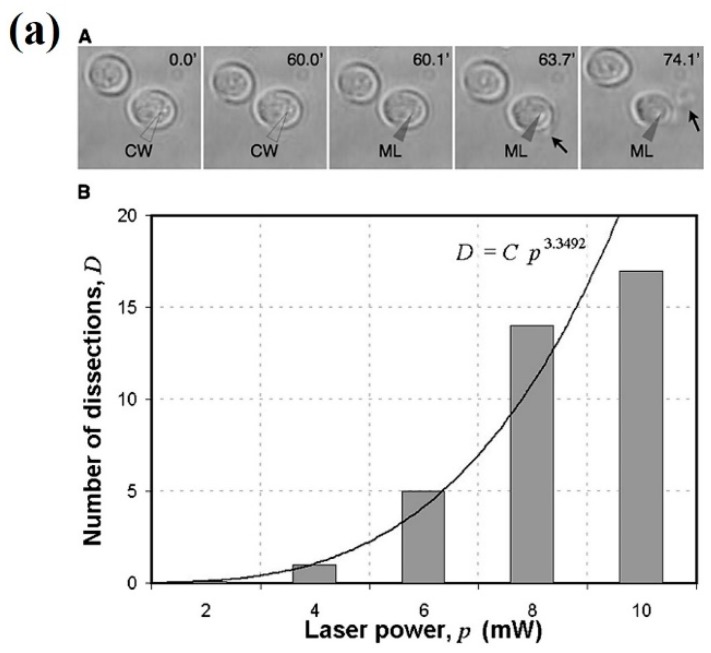
(**a**) Yeast cells laser trapping and dissection. (A) Sequential bright field images of yeast cells with Ti:sapphire laser irradiation. Unfilled and filled triangles indicate the position of the laser focus in CW and femtosecond-pulse operation mode operation, respectively. (B) Number of dissections of yeast cell wall as a function of laser power. Exposure time was fixed to 5 s and 20 cells were irradiated per data point; (**b**) Intracellular organelle extraction and manipulation using the combined technique of optical surgery and trapping. Unfilled and filled triangles indicate the position of the laser focus of the CW and femtosecond-pulsed Ti:sapphire laser, respectively. Black arrows indicate targeted intracellular organelle. Reprinted with permission from [105].

**Figure 17 micromachines-10-00507-f017:**
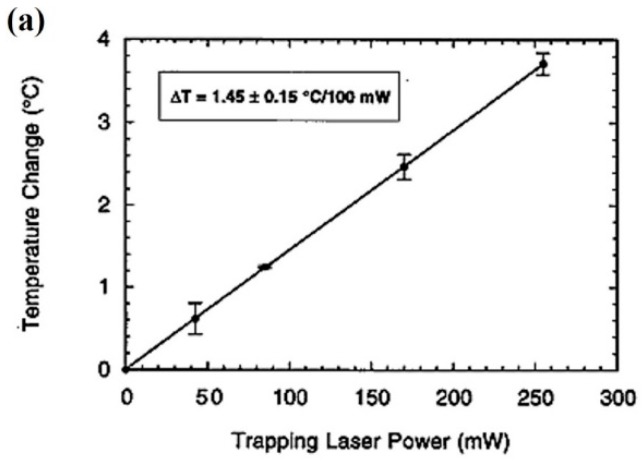
(**a**) Relationship between induced temperature change of aqueous liposomes and laser power of the incident NIR (1064 nm) trapping beam. Error bars represent the standard deviation for multiple measurements made at a given power level. Reprinted with permission from [106]; (**b**) Summary of the experimentally determined values for the laser-induced heating of different microparticles in glycerol and water [108]. The results for fitting the power spectra of thermal motion of trapped beads to the viscous drag experiments are shown. Reprinted with permission from [108].

**Figure 18 micromachines-10-00507-f018:**
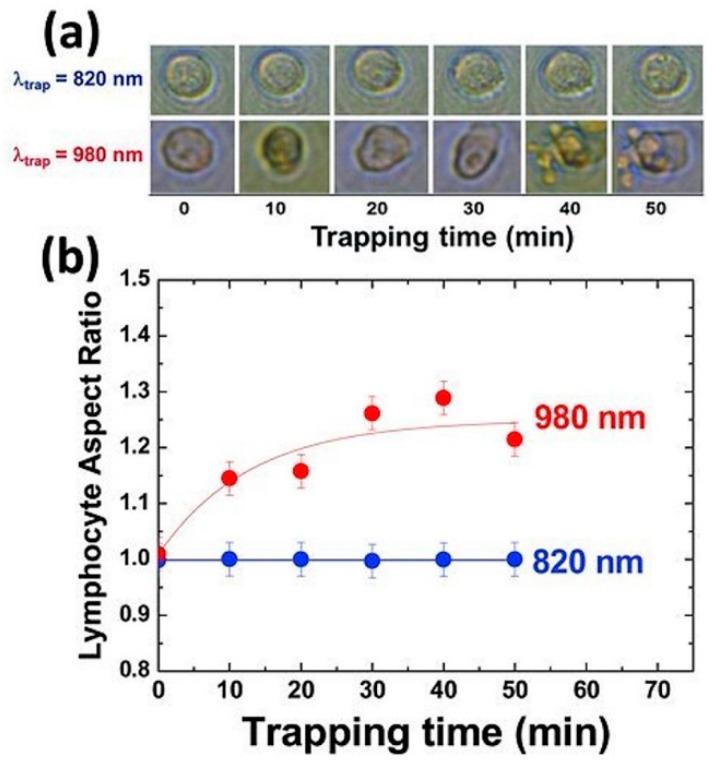
(**a**) Optical transmission images of a single trapped lymphocyte as obtained for different trapping durations. Results obtained for 980 and 820 nm laser trapping wavelengths are included; (**b**) Time evolution of lymphocyte’s aspect ratio during 820 and 980 nm optical trapping. Dots are experimental data and solid lines are guides for the eyes. Reprinted with permission from [110].

**Figure 19 micromachines-10-00507-f019:**
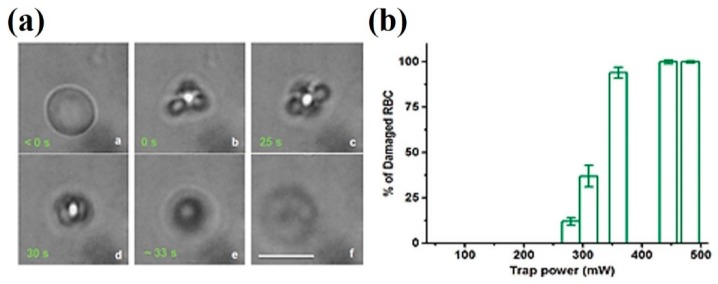
(**a**) The image frames showing sequence of changes experienced by a trapped RBC in phosphate buffer at approximately 360 mW laser power: (**a**) before getting trapped, (**b**) just trapped, (**c**–**e**) changing appearance of the cell in trap at 25 seconds (**c**), 30 seconds (**d**) just before ejection from the trap at 33 seconds (**e**). (**f**) The cell has been just ejected from the trap. Scale bar: 10 μm; (b) The percentage of RBCs damaged at different trap laser powers. Measurements were carried out for trap laser powers of 64, 108, 172, 279, 311, 357, 444, 483 mW. Up to 279 mW, no cells were damaged. For laser powers >357 mW, most RBCs suffered damage when trapped. The errors have been estimated over 5 samples. Reprinted with permission from [113].

**Figure 20 micromachines-10-00507-f020:**
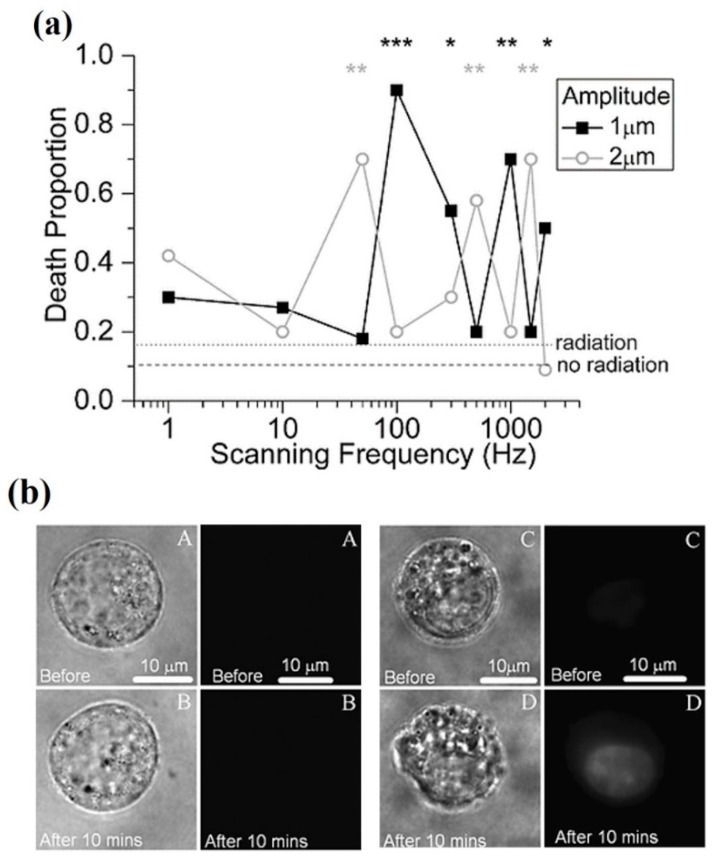
(**a**) Death proportion versus scanning frequency curves with the scanning amplitudes of 1mm (filled squares) and 2mm (empty circle). The dashed and dotted lines represent the death proportions due to the incubation environment and static radiation, respectively. Stars indicate the *P* value: *<0.05, **<0.01, ***<0.001.; (**b**) optical (left) and fluorescence (right) micrographs (A) before and (B) after 10 min of static radiation and (C) before and (D) after 10 min of scanning at 100 Hz and 1mm amplitude. Reprinted with permission from [132].

**Table 1 micromachines-10-00507-t001:** Relevant examples of linear photodamage.

Wavelength	Laser Source (all CW)	Power	Biological Model	Remarks	Reference
514.5 nm	Ar ion	10–100 mW	*E. coli*	Earliest report on biological damage	[5]
405 nm	Diode	75 nW/pixel	HeLa	Assessment of DNA damage response	[44]
405 nm	Diode	1–8 mW	HeLa; mouse fibroblasts; human fibroblasts	Analysis of DNA damage by visible wavelengths	[49]
488 nm	Ar ion				
561 nm	DPSS				
633 nm	He-Ne				
473 nm	Diode	1 mW	HeLa	Autophagy photo-activation	[50]
543 nm					
650 nm					
785 nm	Ti:sapphire	3–9 mW	RBC	Kinetics of hemoglobin photo-degradation	[52]

CW: continuous wave emission; DPSS: diode-pumped solid state laser; RBC: red blood cell.

**Table 2 micromachines-10-00507-t002:** Selected examples of photodamage due to direct optical excitation of ^1^O_2_.

Wavelength	Laser Source (all CW)	Power	Biological Model	Remarks	Reference
750–850 nm	Ti:sapphire	60–120 mW	NC-37	Analysis of DNA damage by NIR wavelengths	[42]
1064 nm	Nd:YAG	60–240 mW			
1270 nm	Raman fiber	100–400 mW	MCF-7	Cell death induction	[67]
1268 nm	Q-dot diode	200–500 mW	HaCaT; primary keratinocytes; HeLa	Cell death induction	[68]
700–990 nm	Ti:sapphire	88–176 mW	CHO	Assessment of cell cloning capacity	[71]
1064 nm	Nd:YAG	88–176 mW			
1064 nm	Nd:YAG	3–30 mW	*E. coli*	Cell mobility and damage	[75]
1064 nm	Nd:YAG	30–230 mW	DNA molecules	Analysis of molecular structural damage	[79]
1070 nm	Yb fiber	0.7–2.6 mW	*S. cerevisiae*	Study of cell growth and division	[80]
1064 nm	Nd:YAG	19–95 mW	*S. cerevisiae*	Study of cell growth and division	[85]
700–840 nm	Ti:sapphire	130 mW	PtK_2_	Assessment of chromosomal abnormalities	[86]
700–850 nm	Ti:sapphire	360 mW	*C. elegans*	Study of heat shock protein induction	[87]
730–800 nm	Ti:sapphire (pulsed fs)	91.5 mW (average power)	HeLa	Necrotic cell death induction	[88]
740–760 nm	Ti:sapphire	88–176 mW	CHO	Induction of “giant” cells	[91]

CW—continuous wave emission; Q-dot—quantum dot.

**Table 3 micromachines-10-00507-t003:** Selected examples of photodamage due to non-linear optical excitation of biological systems.

Wavelength	Laser Source (all pulsed)	Energy/Power	Biological Model	Remarks	Reference
308 nm	XeCl excimer (ns)	525 Jm^−2^ (flux/pulse)	Human lymphocytes	Analysis of DNA damage by UVA-VIS wavelengths	[41]
312–318 nm	Doubled dye (ns)	9 Jm^−2^ (flux/pulse)			
340–640 nm	Dye (ns)	260–460 Jm^−2^ (flux/pulse)			
337 nm	Nitrogen (ns)	4-40 nJ/pulse	HeLa	Analysis of DNA damage at several wavelengths	[44]
532 nm	Nd:YAG (ns)	31 nJ/pulse			
532 nm	Nd:YVO_4_ (ps)	44 nJ/pulse			
800 nm	Ti:sapphire (fs)	0.47 nJ/pulse			
>750 nm	Ti:sapphire (fs)	10–70 mW (average)	HeLa	Autophagy induction	[50]
1064 nm	Nd:YAG (ns)	0-500 μJ/pulse	CHO; human spermatozoa	Cell viability assessment	[70]
750–800 nm	Ti:sapphire (fs)	70–88 mW (average)	CHO; human spermatozoa	Cell viability assessment	[90]
1064 nm	Nd:YAG (CW)	20–400 mW	CHO; human spermatozoa	Cell viability assessment	[95]
840 nm	Ti:sapphire (fs)	7–75 mW (average)	Bovine adrenal chromaffin cells	Study Ca^2+^ uptake and degranulation reaction	[98]
800 nm	Ti:sapphire (fs)	1.8–20 mW (average)	RBC	Plasmatic membrane collapse	[103]

UVA—ultraviolet A; VIS—visible optical wavelengths (400–700 nm); CW—continuous wave emission; RBC—red blood cell.

**Table 4 micromachines-10-00507-t004:** Examples of thermal damage in NIR optical traps.

Wavelength	Laser Source (all CW)	Power	Biological Model	T Increase (°C/100 mW Optical Power)	Reference
1064 nm	Nd:YAG	4–160 mW	*E. coli*	“*estimated to be several degrees Centigrade*.”	[6]
			*S. cerevisiae*		
			RBC		
			Spirogyra		
			Protozoa		
1064 nm	Nd:YAG	45–255 mW	Liposomes	1.45 ± 0.15 °C	[106]
			CHO	1.15 ± 0.25 °C	
985 nm	Yb fiber	55 mW	Water	7.3 °C	[107]
750/920 nm	Ti:sapphire	15–200 mW	Jurkat	0 °C/5 °C	[110]
808/980 nm	Diode			0 °C/9.9 °C	
1090 nm	Yb fiber			4.9 °C	
1064 nm	Nd:YAG	64–1150 mW	RBC	Unspecified	[113]

CW—continuous wave emission; RBC—red blood cell.

**Table 5 micromachines-10-00507-t005:** Strategies to reduce damage in optical traps.

Strategy		Biological Model	Remarks	References
NIR wavelengths	810–860 nm	*E. coli*	Wavelengths in the biological window, avoiding molecular oxygen absorption bands.	[72,87,88,92,110]
		HeLa		
	940–960 nm	*C. elegans*		
Spatial light patterns	Optical beam patterning	-	Bessel beams, Airy beams, Laguerre-Gaussian beams.	[145]
	Optical funnels	Yeast cell	Annular beams produces hollow trapping light cone	[146]
	Indirect particle trap	Yeast cell	Trapped inorganic particle acts upon biological sample	[147]
Exposure management	Scanning beam	D. melanogaster	Reduce exciting beam dwelling time and/or introduce “pulsing” at 1–100 kHz to reduce photodamage	[149]
	Pulsing exciting beam	HEK cells		[150]
ROS quenchers	NaN_3_	DNA, HeLa, MCF-7	Deactivates ^1^O_2_	[76,79,88]
	α-tocopherol	HaCaT, primary keratinocytes, HeLa	Deactivates ^1^O_2_, reacts with ROS	[68,88]
	BSA	MCF-7	Reacts with ROS	[67]
	Ascorbic acid	DNA	Reacts with ROS	[79]
Oxygen depletion	GODCAT	DNA, E. coli	Enzymatic mixture consumes local oxygen	[79,153]
	POC	-	Consumes oxygen but does not affect pH	[154]
Active thermal control	CW laser pulsing	Water	Sample is exposed to light for short periods	[156]
	Heat sinks/NIR laser-driven convective flows	Water/heavy water	Active thermal sinks to dissipate heat; or NIR laser mildly heating to induce convection and cool fluid entrainment.	[157]
	Microfluidic chambers	Water/Jurkat cells	Micro-flows to increase heat dissipation.	[158]

ROS—reactive oxygen species; NaN_3_—sodium azide; BSA—bovine serum albumin; GODCAT—glucose oxidase-catalase; POC—pyranose oxidase-catalase.

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
