# Peer review of "Optical Tweezers: Phototoxicity and Thermal Stress in Cells and Biomolecules"

_micromachines, 2019, doi:10.3390/mi10080507_

Round 1

Reviewer 1 Report

This paper presents a comprehensive survey of the literature on the damages incurred by optical tweezers on cells and biomolecules, with an emphasis on phototoxicity. It is a well-written paper that provides an up-to-date review of all the experiments that have been conducted to characterize the damages. Some useful observations are also made on how to potentially minimize or reduce these damages, and even make use of them for certain applications such as cell dissection and cell fusion. 

Overall, this is a very useful paper that would be of value to the optical manipulation community, particularly those working on biophysical applications. The quality of the paper could be further enhanced by addressing these two comments:

- While the paper is definitely understandable and readable, it can benefit from a thorough checking of grammatical and/or typographical errors. Certain phrases and/or sentences can also be re-written to avoid awkwardness.

- In lieu of some of the text, it would be useful to include a Table summarizing the key literature-based findings and author-made observations on the damage sources and limitation approaches, respectively. 

Reviewer 2 Report

This manuscript, a quite timely and well-written review, provided an overview of phototoxicity and thermal stress in optical tweezer . Indeed, we can call it photodamage which is caused by the light-matter interaction.  Before this manuscript accepted by micromachines, here are a couple of comments for the author to  address.

 For the title of this review, targeting phototoxicity and thermal stress of optical tweezer in live cells.   However,  I suggested that the author re-organize the structure  of this manuscript so that it can teach the field how to choose appropriate parameters to do optical tweezer  well in living cells.    

(1) As live cells are illuminated by focused laser to trap particle and confocal (multiphoton) image. The cell will be disturbed by light, including photon damage, photon toxicity and photon bleaching.  This will cause cell unhealthy, damage, melting and even death.

(2) Also, we should carefully consider which kinds parameters should be taken in account.

  Wavelength, laser intensity average power, illumination time, focus region size, depth dependence

(3) For optical tweezer, the photon damage in different regions (illumination cone, and focal region) should be discussed for both, with CW laser and  ultra-fast laser cases. There are lots of open access codes calculating power distribution  in cells  or tissues using Monto Carlo methods . It  can provide the thermal/ heating effect estimation in different regions.    

(4) The photodamage is also cell type depending. The manuscript can also have a discuss about it.

(5) As the photodamage in live cells is power dependence. The threshold of laser power, illumination time can also be discussed. This should be a good guideline to optical tweezer users.

Other suggestions:

I suggest the author make some tables to compare the parameters of  the laser which cause the live cells damage, stress, toxicity ,melting and even death.
